# Synthesis of vancomycin fluorescent probes that retain antimicrobial activity, identify Gram-positive bacteria, and detect Gram-negative outer membrane damage

Bing Zhang [1], Wanida Phetsang[1], M. Rhia L. Stone[1], Sanjaya Kc[1], Mark S. Butler[1], Matthew A. Cooper [1], Alysha G. Elliott [1], Urszula Łapińska[2,3], Margaritis Voliotis[2,4], Krasimira Tsaneva-Atanasova [2,4,5,6], Stefano Pagliara [2,3] & Mark A. T. Blaskovich [1✉]

Antimicrobial resistance is an urgent threat to human health, and new antibacterial drugs are desperately needed, as are research tools to aid in their discovery and development. Vancomycin is a glycopeptide antibiotic that is widely used for the treatment of Gram-positive infections, such as life-threatening systemic diseases caused by methicillin-resistant *Staphylococcus aureus* (MRSA). Here we demonstrate that modification of vancomycin by introduction of an azide substituent provides a versatile intermediate that can undergo copper-catalysed azide—alkyne cycloaddition (CuAAC) reaction with various alkynes to readily prepare vancomycin fluorescent probes. We describe the facile synthesis of three probes that retain similar antibacterial profiles to the parent vancomycin antibiotic. We demonstrate the versatility of these probes for the detection and visualisation of Gram-positive bacteria by a range of methods, including plate reader quantification, flow cytometry analysis, high-resolution microscopy imaging, and single cell microfluidics analysis. In parallel, we demonstrate their utility in measuring outer-membrane permeabilisation of Gram-negative bacteria. The probes are useful tools that may facilitate detection of infections and development of new antibiotics.

[1] Centre for Superbug Solutions, Institute for Molecular Bioscience, The University of Queensland, Brisbane, QLD 4072, Australia. [2] Living Systems Institute, University of Exeter, Stocker Road, Exeter EX4 4QD, UK. [3] Biosciences, University of Exeter, Stocker Road, Exeter EX4 4Q, UK. [4] Department of Mathematics, University of Exeter, Stocker Road, Exeter, UK. [5] EPSRC Hub for Quantitative Modelling in Healthcare, University of Exeter, Exeter EX4 4QJ, UK. [6] Department of Bioinformatics and Mathematical Modelling, Institute of Biophysics and Biomedical Engineering, Bulgarian Academy of Sciences, 105 Acad. G. Bonchev Street, 1113 Sofia, Bulgaria. ✉email: m.blaskovich@uq.edu.au

Vancomycin **1**, a glycopeptide antibiotic, is commonly used to treat infections caused by multidrug-resistant Gram-positive bacteria[1]. Vancomycin was introduced to clinical use in the late 1950s, though widespread use was delayed for a number of years due to the toxicity of initial crude preparations[2]. Vancomycin inhibits the growth of bacteria by binding to the terminal D-Ala-D-Ala dipeptide of the peptidoglycan precursor Lipid II and prevents Lipid II from reacting with either trans-peptidases (which catalyse the cross linking step) or transglyco-sylases (which catalyse extension of sugar chains)[3]. Compared to other antibiotics, resistance to vancomycin took a long time to develop, as it was nearly 30 years before vancomycin-resistant enterococcal (VRE) infections were reported in the mid- and late-1980s[4]. Resistance to glycopeptide antibiotics, as found in *Enterococci*, mostly results from expression of the resistance gene clusters: *vanA* and *vanB*. These clusters encode enzymes that produce a modified peptidoglycan precursor D-Ala-D-Lac, instead of D-Ala-D-Ala, reducing the binding affinity of glycopeptides[5,6]. Vancomycin is generally considered as the first-line treatment for serious systemic methicillin-resistant *Staphylococcus aureus* (MRSA) infections[7]. However, reduced susceptibility of MRSA to glycopeptide antibiotics, known as glycopeptide-intermediate *S. aureus* (GISA) (vancomycin minimum inhibitory concentration (MIC) = 4–8 µg/mL compared to MIC ≤ 2 µg/mL for susceptible strains), was first observed in Japan in 1996[8]. GISA strains possess a thicker cell wall compared to glycopeptide-susceptible strains. This thickened cell wall increases the number of 'decoy' binding sites for glycopeptides, leading to trapping of glycopeptide molecules so that fewer glycopeptide molecules can reach the target site at the cytoplasmic membrane[2,9]. The first case of vancomycin-resistant *S. aureus* (VRSA) was reported in 2002 (vancomycin MIC ≥ 16 µg/mL), using the same Lipid II modification as VRE[10,11], though fortunately very few VRSA cases have been reported so far. Overall, given the increasing prevalence of GISA and VRE strains, the effectiveness of vancomycin to treat Gram-positive bacterial infectious diseases is under threat[2].

Technologies used to visualise cellular structure and dynamics in living cells enable scientists to understand the interaction and function of biomolecules within complex biological systems. In particular, understanding the cellular complexity of bacteria, and how antibiotics interact with bacteria, is important in developing new strategies to combat antibiotic-resistant bacteria. A mainstay of bacterial assays assessing antibiotic mechanisms of assay, particularly for those active against Gram-negative bacteria, are the fluorescent dyes that assess membrane damage. *N*-phenyl-1-naphthylamine (NPN) is an uncharged lipophilic probe with low fluorescence quantum yield in an aqueous environment, which becomes fluorescent when partitioned in the hydrophobic environment of a lipid membranes. It is widely used to assess damage to the outer membrane (OM) of Gram-negative bacteria, which normally prevents penetration of NPN[12], but if weakened allows intercalation of NPN into the OM phospholipid inner leaflet and the cytoplasmic membrane, causing increased fluorescence. NPN is often used in combination with dyes that stain the nucleic acids of cells, such as SYTOX® Green, which is impermeant to live cells, with staining being indicative of damage to both the outer and inner membranes[13]. Propidium iodide (PI) is another cell-impermeant dead cell indicator which exhibits enhanced fluorescence upon binding to DNA or RNA, and is now widely used to detect dead cells within a population of bacteria[14].

Fluorescent probes derived from antibiotics have potential advantages over other dyes, as their mechanism-specific binding may provide greater insight into the interaction and function of antibiotics with bacterial cells[15]. To increase the likelihood that molecular probes accurately mimic the mechanistic activity of the parent antibiotic, it is important to ensure that they retain similar biological activity. Fluorescent vancomycin analogues, where vancomycin has been conjugated with BODIPY FL (commercially available from Molecular Probes/Invitrogen/ThermoFisher Scientific as BODIPY™ FL Vancomycin, Cat No. V34850[16–24]), fluorescein[25–29], rhodamine[30,31], Alexa Fluor 532[32], and a near-infrared dye IRDye 800CW[33] have been utilised in investigations into localisation and mode of action[17,18,20,21,24,25,27,28,31,32], biofilm penetration[19], resistance mechanisms[16], and diagnosis[22,26,30,33]. Surprisingly, not all derivatives have had antimicrobial activity reported. For those that have, both fluorescein- and BODIPY-linked vancomycin were orders of magnitude less active than vancomycin against *B. subtilis* (vancomycin MIC = 0.13 µg/mL; fluorescein-VAN MIC = 20 µg/mL; BDP-VAN MIC = 2.5 µg/mL). Notably, the fluorescein derivative, containing a large and negatively charged fluorophore, showed much less inhibitory activity than the vancomycin probe with the small fluorophore BODIPY[25]. This is likely due to the composition of the cell wall of Gram-positive bacteria, formed from proteins, peptidoglycan and teichoic acids (TAs). TAs are divided into two classes: the wall teichoic acids (WTAs), which are covalently linked to peptidoglycan, and the lipoteichoic acids (LTAs), which are attached to the head groups of membrane lipids. TAs have negative charges (phosphate groups)[34–36], which potentially repel the negatively charged fluorescein. Developing a set of vancomycin probes with different colour fluorophores that are readily prepared from a common intermediate, and which retain antibacterial potency similar to the parent antibiotic, would be an important addition to the current suite of molecular probes. Indeed, the probes that are described in detail in this report have already been applied by collaborators to assess changes in Gram-negative bacterial membrane permeability, including confirmation of the mode of action of the potentiator compound PBT2, used in combination with tetracycline-class antibiotics against drug-resistant *Acinetobacter baumannii*[37].

## Results and discussion

**Design of azido-vancomycin**. The glycopeptide antibiotic vancomycin can be modified at several regions that do not directly interfere with binding of vancomycin to its Lipid II target, which depends mainly on interactions with the heptapeptide backbone. The C-terminal carboxy group, primary and secondary amine groups, and hydroxyl and phenolic groups, have all been functionalised[1,38,39]. For this study, vancomycin was modified at the C-terminal carboxy group, as previous studies and our own extensive modifications at this position[40] indicate that this site does not interfere with binding to Lipid II, nor with vancomycin dimerisation.

We have previously reported on the utility of azide-functionalised antibiotics and the Cu-catalysed azide−alkyne cycloaddition (CuAAC) reaction for linking antibiotics to fluorophores[41–45]. The CuAAC reaction is compatible with the multiple unprotected functional groups presented in antibiotics, in this case the many amine, hydroxyl, and amide groups on vancomycin. Other groups have reported successful CuAAC reactions with glycopeptides, including vancomycin, for the synthesis of modified glycopeptides[46–52], glycopeptide dimers[53,54], and glycopeptide fluorescent probes[31]. Therefore, we applied this strategy to the preparation of vancomycin analogues by synthesising three azide-derivatised vancomycins **2**–**4** containing different linkers (Fig. 1). We have previously reported on the preparation of vancomycin-nanoparticle conjugates by reacting the azide-functionalised vancomycin with alkyne-functionalised nanoparticles: these demonstrated enhanced binding affinity to bacteria and caused permeabilisation

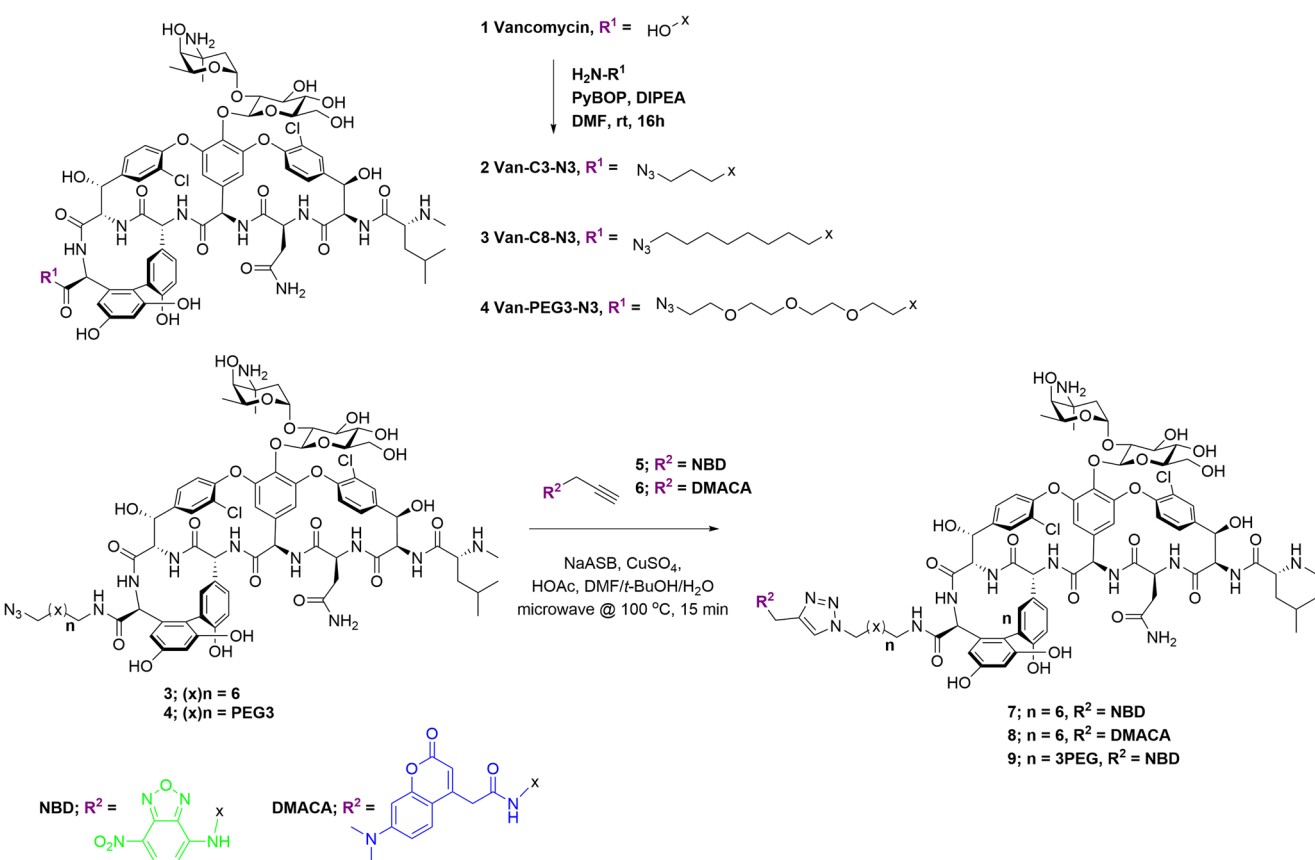

**Fig. 1 Synthesis of azido-vancomycin and fluorescent vancomycin derivatives 7–9.** The *C*-terminus of vancomycin is amidated with amino-azido linker groups using standard peptide coupling reagents. The resulting azides **3** and **4** are then coupled with alkyne-derivatised fluorophores, NBD (green) or DMACA (blue), using the Cu-catalysed azide−alkyne cycloaddition reaction.

of the bacterial cell wall[55]. We are also investigating vancomycin-functionalised magnetic nanoparticles as capture probes for the development of a rapid bacterial diagnostic for blood- and other biological fluid-based infections[56].

**Antimicrobial activity of azido-vancomycin 2–4.** As expected, the azide-vancomycin derivatives **2–4** retained antimicrobial activity against a representative section of Gram-positive bacterial strains including ATCC reference strains and clinical isolates of *S. aureus*, *Streptococcus pneumoniae*, *Enterococcus faecalis*, and *Enterococcus faecium*, with slight improvements in potency in some cases (Table 1). Azide-vancomycin **3** containing the octyl (8-C) linker was the most active, with MICs against most strains consistently two- to four-fold more potent than vancomycin (e.g. 0.5 μg/mL against MRSA, 32 μg/mL against VRE, 2 μg/mL against GISA compared to 1–2 μg/mL, ≥32 μg/mL, and 8 μg/mL respectively for vancomycin; our previous research into vancomycin analogues has shown lipophilicity at this position improves potency[39]). Both azido-vancomycin **2**, with a shorter propyl (3-C) linker, and azido-vancomycin **4**, with a more hydrophilic but longer polyethyleneglycol (PEG$_3$) chain, showed comparable activity to vancomycin.

**Conjugation with fluorophores.** Fluorescent vancomycin probes were readily prepared by a single step synthesis with alkyne-derivatised fluorophores, using azido-vancomycins **3** and **4** as representative azides with different linker properties (both length, and hydrophobicity). To avoid undesirable steric or electronic interaction of the fluorophores with charged cell walls and minimise disturbance to antimicrobial activity, the fluorophores

7-**ni**tro**b**enzofurazan (NBD) (green) and 7-(**dim**ethyl**a**mino)-**c**oumarin-4-**a**cetic acid (DMACA) (blue) were selected, due to their comparatively low molecular weight and minimal (or positive) electronic charges (positive charge less likely to repel negatively charged bacterial surface). We have previously reported on the functionalisation of other antibiotic classes using alkyne-functionalised NBD **5** and DMACA **6**[41–45].

Initially, CuAAC reactions of azido-vancomycin with the alkyne fluorophores gave poor yields of products **7–9** (Fig. 1). In previous literature reports, Nigam[50] synthesised CRAMP-vancomycin via a CuAAC reaction (CuSO$_4$·5H$_2$O and sodium ascorbate (NaASB)) under microwave irradiation. CuAAC reaction under microwave irradiation has also been reported for the synthesis of a vancomycin hybrid[49]. However, this approach was not successful for our constructs, even when tris[(1-benzyl-1*H*-1,2,3-triazol-4-yl) methyl]amine (TBTA) ligand was also added: the reaction was sluggish and isolation of the product was not possible. Alternative copper sources such as Cu(OAc)$_2$, CuI, and Cu(CAN)$_4$PF$_6$ were also evaluated, but with little success. Finally, reaction with CuSO$_4$·5H$_2$O (up to 0.5 equivalents) and sodium ascorbate at 40 °C in DMF/H$_2$O (1:9) for 20 h gave good yields of the conjugation products[50]. Vancomycin is known to chelate copper, which may partially explain the poor yields of some of these reactions[57].

The reaction was further optimised by using a solvent mixture of DMF/*t*-BuOH/H$_2$O and adding acetic acid to accelerate the reaction, based on reports that acetic acid as a proton source accelerates the conversion of the C-Cu bond-containing intermediates (Cu(I) acetylide and 5-cuprated 1,2,3-triazole) into product, leading to reduction of by-products[58,59]. Given that

**Table 1 Antimicrobial activity of vancomycin derivatives against Gram-positive bacteria[a].**

| Compound | MIC (µg/mL) | | | | | | | | | | | | |
|---|---|---|---|---|---|---|---|---|---|---|---|---|---|
| | S. aureus | | | | | | | | S. pneumoniae | E. faecalis | E. faecium | | |
| | ATCC 25923 Control | Clinical isolate MRSA | ATCC 43300 MRSA | NRS 17 GISA | NRS 1 GISA, MRSA | Clinical isolate MRSA, DRSA | VRS 3b VRSA | VRS 10 VRSA | ATCC 700677 MDR | ATCC 29212 Control | Clinical isolate VanA | ATCC 35667 Control | ATCC 51559 MDR-VanA |
| 1 Vancomycin | 2 | 2 | 1 | 8 | 8 | 2 | 64 | >64 | 2 | 4 | 32 | 0.5 | >64 |
| 2 Van-3C-N₃ | N/A | 2 | 2 | 4 | 4 | 2 | >64 | >64 | 2 | 2 | >64 | 1 | >64 |
| 3 Van-8C-N₃ | N/A | 0.5 | 0.5 | 2 | 2 | 0.5 | 16 | 64 | 0.5 | 0.5 | 32 | 0.25 | 32 |
| 4 Van-3PEG-N₃ | N/A | 4 | 2 | 8 | 8 | 4 | >64 | >64 | 4 | 0.5 | >64 | 1 | >64 |
| 7 Van-8C-Tz-NBD | 0.5 | 1 | 0.5 | 2 | 2 | 0.5 | 16 | 16 | 0.5 | 0.5 | 16 | 0.06 | 32 |
| 8 Van-8C-Tz-DMACA | 0.5 | 1 | 0.5 | 2 | 4 | 1 | 32 | 32 | 1 | 0.5 | 16 | 0.125 | 32 |
| 9 Van-3PEG-Tz-NBD | 2 | 4 | 2 | 8 | 8 | 2 | >64 | >64 | 2 | 4 | >64 | 0.5 | >64 |

[a]Full strain descriptions detailed in Table S4. MIC was tested by broth microdilution in non-binding surface (NBS) 96-well micro-titre plates, $n = 4$ (2 independent duplicate experiments).

electronic effects of the alkyne can influence the formation of the Cu(I) acetylide, optimisation of the quantities of $CuSO_4$ and NaASB was evaluated separately for each fluorophore alkyne (**5** required 0.2 eq. $CuSO_4$ and 0.4 eq. NaASB, **6** required 0.5 eq. $CuSO_4$ and 1.0 eq. NaASB). Adding excess catalyst did not improve the outcome of the reaction but increased unidentified by-products. Under the optimised microwave conditions, the CuAAC reactions in DMF/$t$-BuOH/$H_2O$ with $CuSO_4$, NaASB, and HOAc as catalyst were completed within 15 min at 100 °C, providing the fluorescent vancomycins **7–9** in good yield. The excitation/emission wavelengths of Van-8C-Tz-NBD **7** (475/535 nm), when compared to Van-8C-Tz-DMACA **8** (375/480 nm) (Fig. S1), were generally more suitable for fluorescent microscopy as suitable filters are more readily available. Thus, azido-vancomycin **4** was conjugated with the NBD fluorophore **5** to generate another green probe **9**.

**Biological activity and quantification of bacterial staining with fluorescent vancomycin probes.** Both fluorescent probes **7** and **8**, with a C8 alkyl linker, showed improved (approximately two- to four-fold) antibacterial activity against multiple strains, when compared to the parent vancomycin (Table 1), again likely due to lipophilicity. Probe **9**, with the PEG3 linker, retained the most similar antimicrobial activity to the parent vancomycin. We also compared the antimicrobial activity of probes **7, 8** and **9** to the commercially available vancomycin-derived fluorescent probes, Van-FITC (SBR00028, Sigma-Aldrich), and Van-BODIPY (V34850, Invitrogen™). Compared to Van-BODIPY, Probe **7** retained two-fold improved activity against all tested *S. aureus* strains, whereas Van-FITC did not possess antimicrobial activity at the highest concentration tested (32 µg/mL) (Table S3). The chemical stability of Probe **9** was assessed by analytical testing (HPLC) of old stock solutions that were dissolved in $ddH_2O$ and stored at −20 °C for over 3 years. No obvious degradation was evident, with purity >95% as determined by LC-MS using the same method employed to assess purity after synthesis. Furthermore, a one-year-old stock of Probe **9** showed identical antimicrobial activity as the fresh stock, indicating that the probe can be used for at least one year once dissolved (Table S3). For testing of Van-FITC and Van-BODIPY, they were both dissolved in DMSO according to their product manuals, instead of the water used for probes **7–9**. Given that water is less disruptive to bacterial survival than DMSO, the ability to dissolve the new probes in water should cause less experimental interference.

We next assessed the selectivity of the probes at distinguishing Gram-positive from Gram-negative bacteria. Traditionally, this is accomplished by Gram-staining, which relies on uptake and retention of a crystal violet stain to label the thick peptidoglycan layer of Gram-positive bacteria. However, many factors including the intactness and size of bacterial cells will affect the staining accuracy of this commonly used technique and it is not suitable for investigating viable bacterial cells because of a fixation step. Gram-negative-specific fluorescent probes such as polymyxin B-Cy3[60], tetramethylrhodamine-labelled tridecaptin A1[61], and Gram-positive-specific fluorescent probes such as vancomycin-Cy5[62], provide a new approach for labelling live Gram-negative bacteria and Gram-positive bacteria, respectively. In this study, the selectivity of vancomycin probes **7–9** were assessed. Flow cytometry analysis of bacteria labelled with the vancomycin fluorescent probes **7–9** clearly show selective labelling of *S. aureus* but not *E. coli* (Fig. 2a). The Van-DMACA probe **8** was somewhat less selective than the NBD-based probes (selectivity ratio for *S. aureus* over *E. coli* based on fluorescence uptake: **7** = 101:1, **8** = 25:1, **9** = 52:1). To further assess the specific labelling by

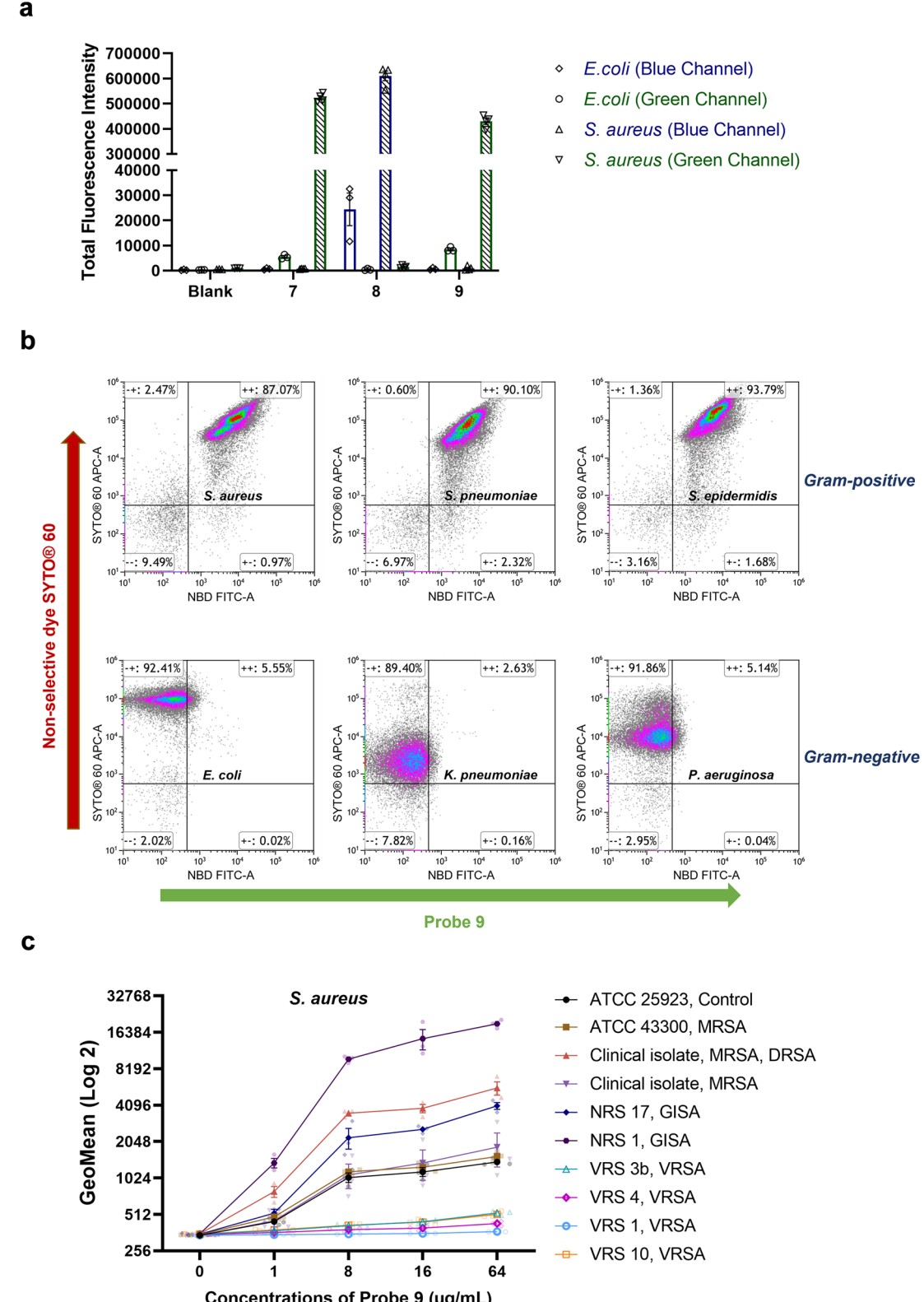

probe **9**, we analysed 3 Gram-positive and 3 Gram-negative bacteria co-stained with the green probe **9** and a non-specific cell permeable red nucleic acid dye SYTO® 60. SYTO® 60 stained all 6 strains while a shift in the FITC channel only appeared for Gram-positive strains, confirming selectivity between Gram-positive and Gram-negative bacteria (Fig. 2b).

We also compared fluorescent uptake of probe **9** by different *S. aureus* strains with varying degrees of vancomycin resistance, with vancomycin-sensitive and GISA strains showing a clear concentration-dependence (Fig. 2c). GISA and daptomycin-resistant strains, resistance phenotypes generally accompanied by thickened cell walls, showed moderately to highly increased

**Fig. 2 Flow cytometry analysis of bacteria labelled with probes 7–9. a** Labelling of *S. aureus* ATCC 25923 and *E. coli* ATCC 25922 with vancomycin probes **7–9** measured using flow cytometric analysis. Bacteria were treated with vancomycin probes at a concentration of 16 μg/mL (37 °C, 30 min), with fluorescence intensity of individual bacteria measured using flow cytometry. Positive events were gated on the histogram to estimate the mean fluorescence intensity (MFI). Total fluorescence intensity (TFI) of the gate was obtained by multiplying the MFI with the number of events on the gate. The data are presented as the mean ± SEM ($n \geq 3$). **b** Labelling of bacteria with the Gram-positive selective vancomycin probe **9** and the non-selective cell permeable dye SYTO® 60 in various species using flow cytometry. Bacteria were treated with vancomycin probes at a concentration of 32 μg/mL (37 °C, 30 min) and SYTO® 60 (5 μM, 10 min on ice), then fluorescence intensity was measured using the flow cytometer. **c** Labelling of *S. aureus* with vancomycin probe **9** measured using flow cytometric analysis. Bacteria were treated with vancomycin probe **9** at different concentrations (37 °C, 30 min), then fluorescence intensity was measured using the flow cytometer. The data are presented as the mean ± SEM ($n = 3$).

levels of fluorescence (e.g. ~4- to ~18-fold increased fluorescence at 64 μg/mL of probe **9** for these strains compared to ATCC 25923). In contrast, VRSA strains showed substantially reduced staining (~6- to ~50-fold decreased fluorescence at 64 μg/mL of probe **9** for these strains compared to ATCC 25923), consistent with their different mechanism of resistance that reduces vancomycin's affinity (Fig. **2**c and Table S1).

We also compared the staining ability of probe **7** and probe **9** to the commercial fluorescent probes using flow cytometry. The Van-BODIPY derivative performed best in staining *S. aureus* with the greatest intensity under the tested conditions. Probes **7** and **9** (as well as the one-year-old stock of Probe **9**) were less effective, while the staining ability of Van-FITC was the worst (Fig. S2a). On the other hand, when incubated with the *E. coli* cells, Van-BODIPY stained a larger population of cells than the other probes, especially at 2 μg/mL, indicating a degree of non-specificity. In contrast, Probes **9** and Van-FITC did not show much staining with *E. coli* at either concentration, while Probe **7** was similar to Probe **9** at 2 μg/mL but had the highest *E. coli* labelling at 16 μg/mL (Fig. S2b).

**Super-resolution confocal microscopy of bacterial staining with fluorescent vancomycin probes**. Super-resolution structured illumination microscopy (SR-SIM) is an essential tool to visualise bacterial cell structures, given their small size (approximately 1 μm) relative to human cells. Thus, *S. aureus* labelled with vancomycin probes **7–9** was fluorescently imaged using SR-SIM (Fig. **3**a). Intense fluorescence at the dividing septum compared to the lateral wall confirmed binding to the nascent peptidoglycan Lipid II. The PEG linker (**9**) and the alkyl linker (**7**) did not lead to obvious differences in visualisation, although probe **7** was approximately 4-fold more active than probe **9** in MIC assays against the Gram-positive panel. The dividing septum ring was even more clearly visible when viewed using 3D-SIM (Figs. **3**b, c and S3 and Supplementary Video). Similar labelling was seen in a range of other Gram-positive species and strains, including MSSA ATCC 25923, MRSA ATCC 43300, VISA NRS 1, VRSA VRS 4, *Staphylococcus epidermidis* ATCC 12228, *Streptococcus pneumoniae* ATCC 33400, *Enterococcus faecium* ATCC 35667 and *Bacillus subtilis* ATCC 6633 (Fig. S4). Notably, despite the relatively low levels of fluorescent uptake indicated by flow cytometry for VRSA, clear septal and cell wall labelling was still visible by microscopy.

Vancomycin probes have been used to study mechanisms of bacterial cell division via experiments including co-labelling with PG synthesis proteins[63], fluorescence microscopy of vancomycin-labelled bacteria to show the dynamics of peptidoglycan assembly in ovococci[17], and investigations of PG biosynthesis in *B. subtilis*[25]. Most bacterial cells grow and divided by building a new cell wall disc called the septum, which is the structure that forms in the middle of the mother cell by invagination of the cell membrane and ingrowth of the cell wall, leading to splitting of the mother cell into two identical daughter cells[64]. SR-SIM fluorescence microscopy successfully showed these stages of cell division, including ingrowth of the new PG, invagination of the cell membrane, and chromosome segregation, when labelled with the vancomycin probes, a membrane-labelling probe, and a nucleic acid-labelling probe, respectively (Fig. **3**d–f).

Airyscan confocal super-resolution imaging was used to compare vancomycin probes **7** and **9**, as well as the commercial probes Van-FITC and Van-BODIPY, for staining of *S. aureus* ATCC 25923 (Fig. S2c). With the exception of Van-FITC, all fluorescent probes showed good peptidoglycan binding ability. However, while Probes **7** and **9** ((as well as one-year-old stock of Probe **9**) showed very strong fluorescence at the cell division septum (the site of Lipid II precursor incorporation into peptidoglycan) compared to the lateral wall, the Van-BODIPY probe appeared to label the cells more uniformly, indicating reduced selectivity of binding to nascent peptidoglycan. More intracellular labelling was also observed with the Van-BODIPY probe, further supporting that Probes **7** and **9** were more specific at labelling nascent peptidoglycan than Van-BODIPY (Fig. S2c). Van-FITC did not show good staining ability, consistent with its poor MIC and flow cytometry staining, with no fluorescence signal observed at 2 μg/mL and less clear labelling than other tested probes (Fig. S2c).

The vancomycin probes are able to spatially discriminate the peptidoglycan layer from the bacterial cell membrane as shown by cross-section measurements (Fig. **3**g), which clearly show the peak of probe intensity outside of the peak intensity of a membrane-selective dye. Thus, vancomycin probes are potentially useful for helping to assess the target sites of other antibiotic probes such as those derived from linezolid and trimethoprim (TMP) with intracellular targets, by providing a reference point for the PG layer. Co-staining of *S. aureus* with the Van-DMACA probe **8** (blue) and trimethoprim-4C-Tz-NBD **17**[42] (green) or linezolid-3C-Tz-NBD **18**[41] (green) showed that the TMP and linezolid probes penetrated within the PG and bacterial cell membranes into the cytosol. Both TMP and linezolid probes generally showed co-localisation with the FM4-64FX (membrane dye) because the membrane dye was quickly endocytosed in *S. aureus* (Fig. **3**h, i). These studies highlight the advantages of having additional colour versions of the antibiotic-fluorophore probes available for use in combination labelling studies.

**Determining membrane permeabilisation in Gram-negative bacteria with fluorescent vancomycin probes: microscopy**. The OM of Gram-negative bacteria is an asymmetric bilayer that contains both phospholipid (inner leaflet) and LPS (outer leaflet). The LPS consists of Lipid A (a glucosamine-based phospholipid), a relatively short core oligosaccharide, and a distal polysaccharide (O-antigen)[65]. Due to the OM preventing access, vancomycin is inactive against Gram-negative bacteria (MIC > 64 μg/mL) (Table **2**). Microscopy of *E. coli* (ATCC 25922, K12 (MB4827)) co-labelled with membrane dye FM4-64FX and vancomycin NBD probe **9** generally showed little evidence of vancomycin binding, as expected (Fig. **4**a, b). However, occasionally cells exhibited

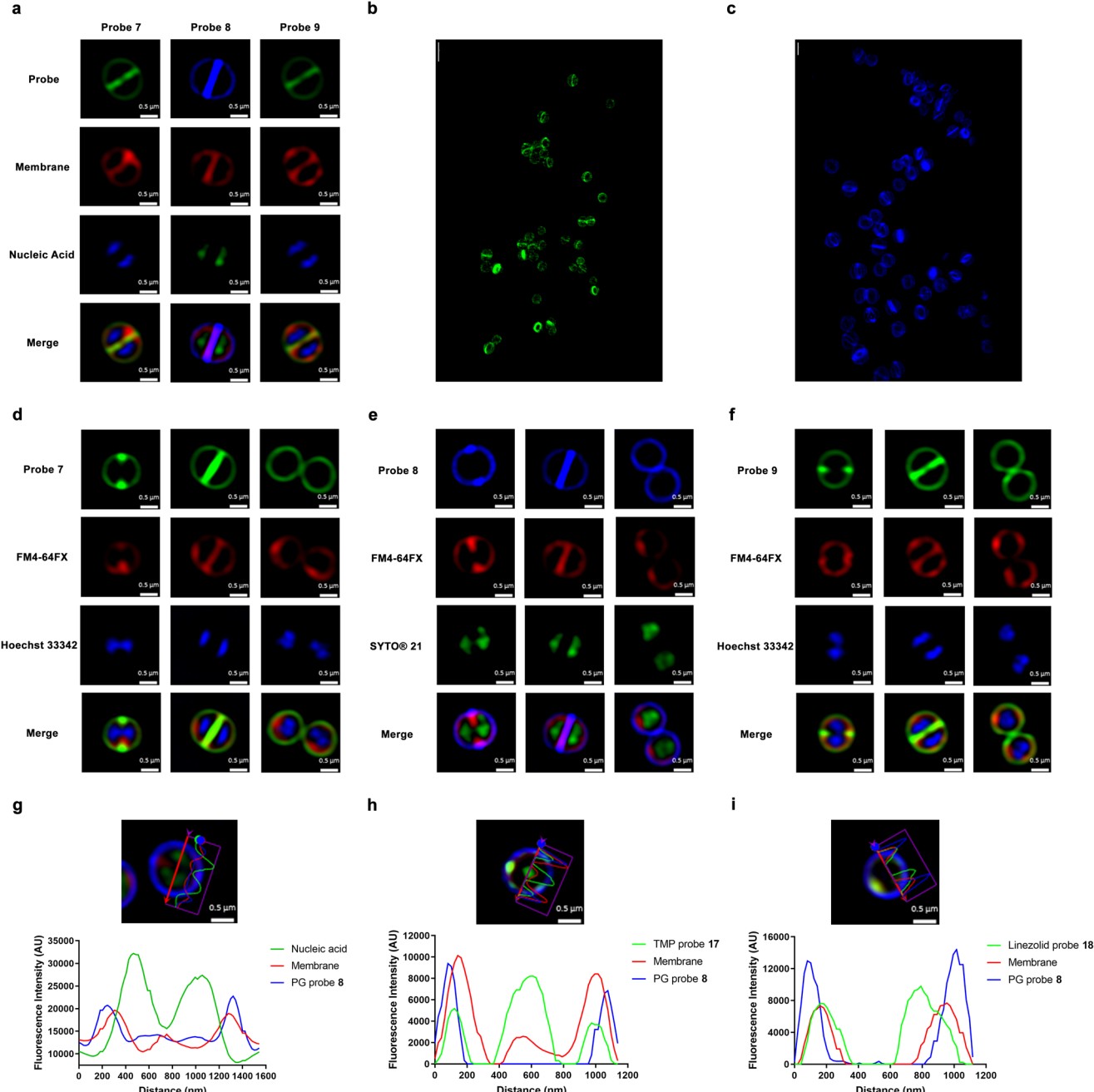

**Fig. 3 High-resolution microscopy of *S. aureus* labelled with probes 7–9. a** SR-SIM fluorescence imaging of *S. aureus* ATCC 25923 stained with probes **7–9**. Bacteria were stained with Hoechst 33342 (blue, nucleic acid) or SYTO® 21 (green, nucleic acid) at 2.5 μM for 5 min on ice, followed by staining of FM4-64FX (red, bacterial membrane) at 5 μg/mL for 2 min on ice. Finally, vancomycin probes (16 μg/mL) were then added to the bacteria, left for 30 min on ice. Scale bar = 0.5 μm. **b** 3D-SIM of probe **7** (green) and **c** 3D-SIM of probe **8** (blue). Scale bar = 2 μm. **d**–**f** SR-SIM fluorescence imaging of *S. aureus* ATCC 25923 cell division with different probes. **d** Green; probe 7, Red; FM4-64FX (bacterial membrane), Blue; Hoechst 33342 (nucleic acid). **e** Blue; probe **8**, Red; FM4-64FX (bacterial membrane), Green; SYTO® 21 (nucleic acid). **f** Green; probe **9**, Red; FM4-64FX (bacterial membrane), Blue; Hoechst 33342 (nucleic acid). Scale bar = 0.5 μm. **g**–**i** Co-labelling of peptidoglycan (PG) labelling vancomycin probe **8** with trimethoprim and linezolid probes. The microscopy image shows SR-SIM fluorescence imaging of *S. aureus* ATCC 25923 with a cross-section measurement of fluorescent intensity in the accompanying graph. **g** Bacteria stained with probes 8 (16 μg/mL, blue, PG), FM4-64FX (5 μg/mL, red, bacterial membrane), SYTOX® green (2 μM, green, nucleic acid). **h** Combination of vancomycin-DMACA 8 (16 μg/mL, blue, PG) with TMP-4C-Tz-NBD probe 17 (64 μg/mL, Green), co-stained with FM4-64FX (5 μg/mL, red, bacterial membrane). **i** Combination of vancomycin-DMACA 8 (16 μg/mL, blue, PG) with linezolid-3C-Tz-NBD probe 18 (64 μg/mL, Green), co-stained with FM4-64FX (5 μg/mL, red, bacterial membrane). Scale bar = 0.5 μm.

strong fluorescence at the dividing septum just before cell division, where presumably the OM becomes 'leaky' as the cell begins to separate. The probe did not show any obvious labelling at the lateral wall. This inspired us to investigate the ability of the vancomycin probes to stain mutant bacteria with impaired OM.

Firstly, vancomycin probe **9** was incubated with permeable membrane mutants of *E. coli*, including a UDP-3-O-((R)-3-hydroxymyristoyl)-*N*-acetyl glucosamine deacetylase (LpxC) mutant MB4902 (deficient in LpxC therefore not efficient Lipid A production) and DC2 mutant CGSC7139 (deficient in

**Table 2 Antimicrobial activity of vancomycin derivatives against *Escherichia coli* strains[a].**

| Compound | E. coli MIC (µg/mL) | | | | | |
|---|---|---|---|---|---|---|
| | ATCC 29522 | MB4827 K12 | DC2 mutant | LpxC mutant | CFT073 | CFT073 *WaaL* mutant |
| Polymyxin B | 0.06 | ≤0.03 | ≤0.03 | ≤0.03 | ≤0.03 | ≤0.03 |
| 1 Vancomycin | >64 | >64 | >64 | 64 | >64 | >64 |
| 7 Van-8C-Tz-NBD | >64 | >64 | >64 | >64 | >64 | >64 |
| 8 Van-8C-Tz-DMACA | >64 | >64 | >64 | >64 | >64 | >64 |
| 9 Van-3PEG-Tz-NBD | >64 | >64 | >64 | 32–64 | >64 | >64 |

[a]Full strain descriptions are detailed in Table S4. MIC was tested in non-binding surface (NBS) 96-well micro-titre plates.

osmoregulation of periplasmic oligosaccharide synthesis), comparing probe uptake to that of the parent strain, K12 (MB4827). Probe **9** was able to strongly label the mutant DC2 *E. coli*, indicating general penetration to the PG layer (Fig. 4c). The *lpxC* mutant *E. coli* also allowed probe **9** to label the PG, with apparently greater penetration to the cytoplasm compared to both control and the mutant DC2 strains (Fig. 4c, d). In MIC assays, both vancomycin and the probe **9** did show weak antimicrobial activity against the *lpxC* mutant *E. coli* (MIC = 64 µg/mL) (Table 2), confirming that LpxC deletion does allow for vancomycin penetration to the PG layer[66–68].

Secondly, because mutation of O-antigen is known to alter antibiotic sensitivities[65,69], an O-antigen mutant *waaL E. coli* with short length LPS (rough LPS mutant), was also stained with probe **9** and compared to the corresponding parent *E. coli* strain (CFT073) containing an intact O-antigen (Fig. 4e, f). In contrast to DC2 and *lpxC* mutants, probe **9** showed a similar lack of labelling in control and mutant O-antigen strains, indicating that O-antigen did not affect the permeability of the probe **9** to the PG under the tested condition (Fig. 4a–f).

Recent studies have suggested that vancomycin is able to inhibit growth of *E.coli* at low temperatures and that cold stress makes *E. coli* susceptible to glycopeptide antibiotics by altering outer membrane integrity[70,71]. To identify if the vancomycin fluorescent probe could detect this membrane damage caused by cold stress, we labelled *E. coli* cells grown at both 37 °C and 15 °C with vancomycin probe **9**. Cells at 37 °C showed minimal labelling, which is consistent with our previous results. In contrast, cells grown at 15 °C showed substantial labelling at both the dividing septum and peptidoglycan (Fig. 4g, h), confirming changes of outer membrane permeability under cold stress.

**Determining membrane permeabilisation in Gram-negative bacteria with fluorescent vancomycin probes: method development**. The ability to visualise OM leakiness with the vancomycin probes triggered an investigation into their utility for quantifying OM damage caused by antibiotics. We developed two methods, using either flow cytometry or a plate reader for detection. *E. coli* ATCC 25922 was treated with 6 membrane-active antibiotics (tachyplesin-1[72], arenicin-3[73], polymyxin B[74], colistin[75], citropin[76] and octapeptin C4[77]) and 3 membrane-inactive antibiotics (gentamicin[78]: aminoglycoside 30S ribosome protein synthesis inhibitor; trimethoprim[79]: dihydrofolate reductase inhibitor, and erythromycin[80]: macrolide 50S ribosome protein synthesis inhibitor) at concentrations in range 0.125–128 µg/mL for 1 h, followed by labelling with the probe **9** (32 µg/mL). With the flow cytometry method (Fig. 4i), pure isopropanol was used as positive control. The fluorescence intensity of vancomycin probe-labelling following treatment with the membrane-active compounds increased with their concentration, while no damage was shown for the antibiotics acting by different mechanisms. For the membrane-active compounds, the level of OM disruption did not necessarily correlate with their MIC (Table S2), but structural classes appeared to exhibit similar trends. The β-hairpin antimicrobial peptides tachyplesin-1[81] and arenicin-3[73] led to increased fluorescence at the lowest concentrations, despite the cyclic lipopeptides polymyxin B and colistin possessing more potent MICs. The less potent cyclic lipopeptide octapeptin C4[77], and the α-helical linear peptide citropin[82] did require substantially higher concentrations before membrane damage was seen. These findings were further confirmed using a Tecan plate reader (Fig. 4j).

**Single-cell microfluidics analysis of OM damage**. Single-cell microfluidics provides an opportunity to bridge the qualitative analysis of tens of cells provided by confocal microscopy, and the bulk population analysis of millions of bacteria provided by flow cytometry and plate reader assays. Using our recently developed single-cell molecular accumulation platform hundreds of individual live bacteria, immobilised within channels in a microfluidic chip, can be monitored over time during the course of exposure to a fluorescent molecule[45,83,84]. Immobilised *E. coli*, exposed to the vancomycin probe **9**, showed individual bacteria sporadically exhibited uptake of the probe, consistent with labelling during cell division (Fig. 4k). In contrast, when the OM permeabiliser polymyxin B was added, a much greater population of bacteria accumulated the probe (Fig. 4l). Next, we used our recently developed mathematical model to capture the phenomenology of drug accumulation in our experiment[85] and inferred the time of onset of fluorescence (denoted $t_0$), the rate of uptake (denoted by the rate constant $k_1$), and steady-state saturation levels (denoted $F_{max}$) describing the accumulation of the probe (Fig. 4m–o). We found that the vancomycin probe started to accumulate significantly faster in individual *E. coli* in the presence of polymyxin B (with a mean $t_0$ of 6750 and 9374 s in the presence and absence of polymyxin B, respectively, ****). Moreover, the rate of uptake of the vancomycin probe was significantly greater in the presence of polymyxin B (with a mean $k_1$ of 0.19 and 0.01 a.u./s$^2$ in the presence and absence of polymyxin B, respectively, ****). Finally, also the steady-state saturation level of the vancomycin probe was significantly higher in the presence of polymyxin B (with a mean $F_{max}$ of 149 and 67 a.u. in the presence and absence of polymyxin B, respectively, ****). Taken together these modelling results affirm the experimental findings presented above.

**Conclusions**
Vancomycin fluorescent probes were designed and synthesised from azido-vancomycin building blocks using the CuAAC reaction. The fluorescent vancomycin probes retained good antimicrobial activity compared to previously reported vancomycin probes, possibly due to the use of small fluorophores with minimal electronic charges, and the procedure allowed for the facile introduction of fluorophores with different colours. We

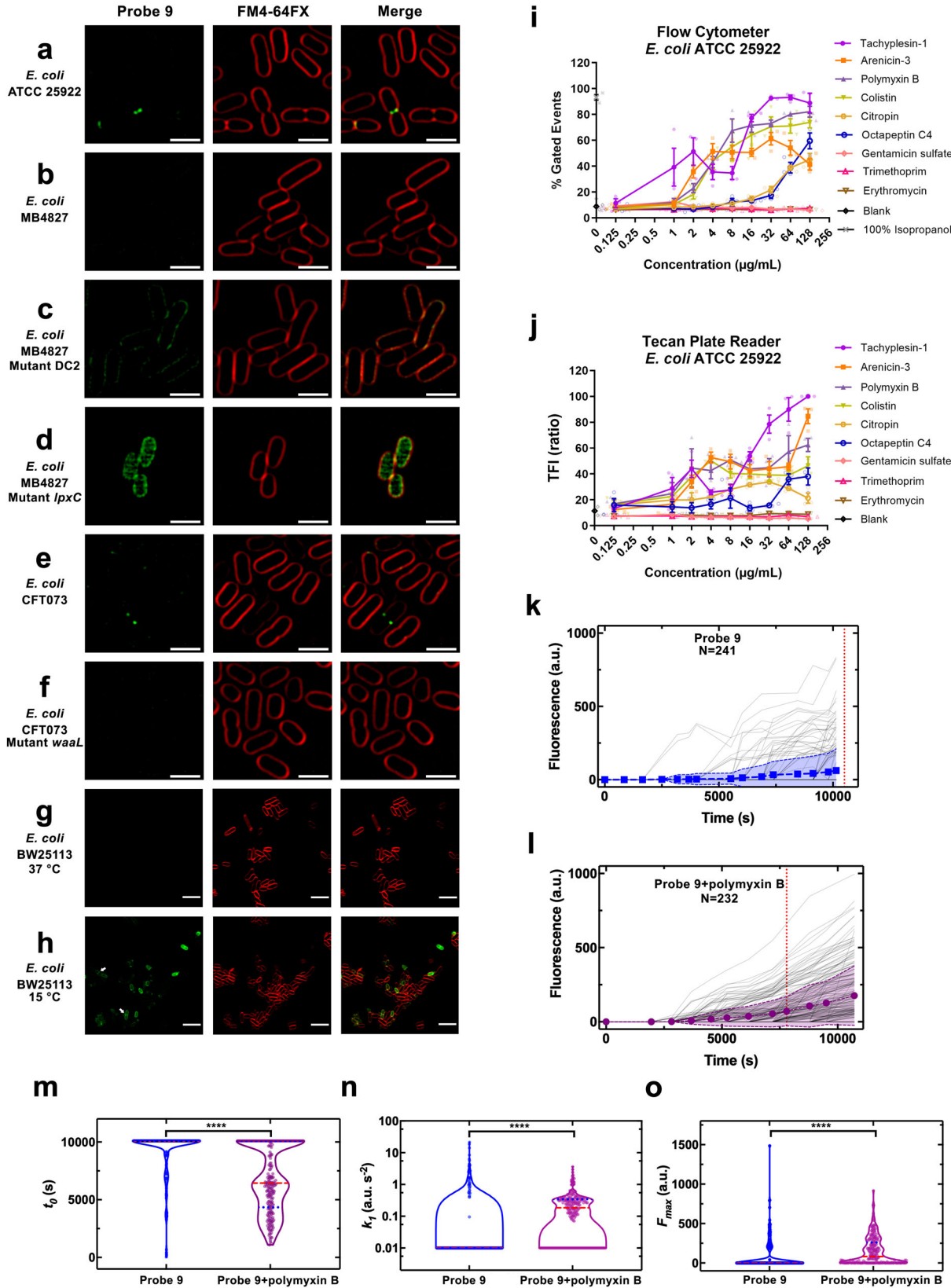

demonstrated the possibility of discriminating Gram-positive and Gram-negative bacteria using the fluorescent probe based on their selective labelling characteristics by flow cytometric analysis. Furthermore, when used for SR-SIM fluorescent imaging of Gram-positive bacteria, the vancomycin probes clearly showed the formation of freshly synthesised peptidoglycan and the probes

were also able to spatially discriminate the peptidoglycan layer from the bacterial cell membrane. Finally, we demonstrated that the vancomycin probe **9** can be used to visualise membrane permeabilisation in Gram-negative bacteria caused by genetic mutations in the bacteria, by temperature stress, or by exposure to chemical compounds, with convenient flow cytometry or plate-

**Fig. 4 Visualisation of membrane permeabilisation in Gram-negative bacteria with fluorescent vancomycin probe 9.** SR-SIM fluorescence imaging of vancomycin probe **9** in *E. coli* strains ATCC 25922 (**a**), K12 (MB4827) (**b**), and CFT073 (**e**), and mutant *E. coli* (DC2 (**c**), *lpxC* (**d**) and *waaL* (**f**)). Bacteria were stained with probe **9** (32 µg/mL, Green) and FM4-64FX (5 µg/mL, red, bacterial membrane). Scale bar = 2 µm. **g**, **h** Visualisation of membrane permeabilisation at different temperatures. Airyscan confocal super-resolution imaging of probe **9** in *E. coli* strain BW25113. Bacteria were stained with probe **9** (32 µg/mL, Green) and FM4-64FX (5 µg/mL, red, bacterial membrane) at 15 °C or 37 °C. Arrows indicate cells labelled with probe **9** at dividing septum and peptidoglycan at 15 °C. Scale bar = 5 µm. **i**, **j** Determination of membrane permeabilisation in *E. coli* (ATCC 25922) using vancomycin probe 9. Bacteria were treated with compounds for 1 h at 37 °C, followed by labelling with the probe Vanco-NBD **9** (32 µg/mL), then left for 30 min at 37 °C. The samples were then measured by flow cytometer or Tecan plate reader. **i** Flow cytometry method. **j** Tecan plate reader method. The data are presented as the mean ± SEM ($n \geq 2$). **k–o** Microfluidic single-cell analysis of probe accumulation in *E. coli*. Accumulation of vancomycin probe **9** in *E. coli* in M9 medium drug milieu at an extracellular concentration of 46 µg/mL in **k** the absence and **l** the presence of polymyxin B at an extracellular concentration of 1 µg/mL delivered to $n = 241$ and $n = 232$ individual *E. coli*, respectively. In both figures, data were collated from biological triplicate and fluorescence values were background subtracted and normalised by cell size. The symbols and shaded areas represent the mean and standard deviation of the corresponding single-cell values. The vertical dotted lines represent the time points at which the median of each dataset became larger than zero. The median remained zero throughout the entire experiments carried out with vancomycin-NBD **9** alone, hence the dotted line has been arbitrarily set at 11,100 s in **k** for comparison purposes only. Distributions of single-cell values for the kinetic parameters (**m**) $t_0$, (**n**) $k_1$ and (**o**) $F_{max}$ describing the accumulation of **9**. The red dashed and blue dotted lines within each violin plot represent the median and quartiles of each data set, respectively. ****$p$ value <0.0001. The fitting algorithm returned convergent transitions for the accumulation data of $n = 46$ (out of the 241 datasets available) bacteria treated with vancomycin-NBD **9** and for $n = 161$ (out of the 232 datasets available) bacteria cotreated with vancomycin-NBD **9** and polymyxin B. $t_0$, $k_1$ and $F_{max}$ values for single-cell accumulation profiles for which the fitting algorithm returned divergent transitions were arbitrarily set to 10,100 s, 0.01 a.u. s$^{-2}$ and 0 for comparison purposes only. These values represent the time at which the experiments with both vancomycin-NBD and the combination of probe **9** and polymyxin B were terminated and the minimum values of $k_1$ and $F_{max}$ measured across the two datasets, respectively.

---

based methods able to quantify the extent of outer membrane damage. Further studies should assess whether these selective staining characteristics are retained across a much larger range of Gram-positive and Gram-negative species and strains, while additional fluorophore colours would help expand the utility of the probes for co-labelling studies. These mechanism-specific methods (as the OM must become permeable to allow access to the peptidoglycan layer) could become a useful addition to the repertoire of tests commonly used to help define the mode of action of new antimicrobial compounds and the effects of bacterial pathway mutations.

## Methods

**General synthetic and analytical procedures.** All materials, unless otherwise noted, were obtained from commercial suppliers and used without further purification. Non-aqueous reactions were conducted under an inert atmosphere of nitrogen. Analytical LCMS was performed on a Shimadzu LCMS 2020 using 0.05% formic acid in water as solvent A and 0.05% formic acid in acetonitrile as solvent B. LCMS conditions (solvent A = H$_2$O + 0.05% formic acid, solvent B = acetonitrile + 0.05% formic acid): Column Zorbax Eclipse XDB-Phenyl, 3.0 × 100 mm, 3.5 µm: Flow: 1 mL/min: Gradient timetable: 0.5 min, 5% B; 8.5 min, 100% B; 2.0 min, 100% B; 0.2 min, 5% B. Biotage Initiator microwave was used for Cu-catalysed azide-alkyne cycloaddition. Column chromatography was performed using silica gel 60 (0.063–0.200 mm), 70–230 mesh ASTM. Agilent 1260 Infinity Preparative HPLC with a G1365D multiple wavelength detector set at $\lambda = 210$ nm and Grace Reveleris X2 chromatography systems were used for compound purification. Commercially available cartridges were used for MPLC chromatography (Reveleris C18 Reversed-Phase 12 g cartridge and column 40 g cartridge), while HPLC purifications used an Agilent Eclipse XDB-Phenyl column 30 × 100 mm, 5 µm particle size. $^1$H (600 MHz) and $^{13}$C (150 MHz) NMR spectra were obtained using a Bruker Avance-600 spectrometer equipped with a TXI cryoprobe. Chemical shifts are reported relative to the residual solvent signals in parts per million ($\delta$) (CDCl$_3$: $^1$H: $\delta$ 7.27, $^{13}$C: $\delta$ 77.2; DMSO-$d_6$: $^1$H: $\delta$ 2.50, $^{13}$C: $\delta$ 39.5). High resolution mass spectrometry (HRMS) was performed on a Bruker Micro TOF mass spectrometer (Ultimate 3000) using (+)-ESI calibrated to HCOONa. The MS/MS was performed on SCIEX X500R QTOF.

Numbering of atoms of vancomycin for NMR analysis is given in Fig. S5, with the additional numbering for the probes **7–9** given in Fig. S6. The $^1$H, $^{13}$C, COSY and HSQC NMR spectra of the parent vancomycin antibiotic are provided as a reference for comparison in Figs. S7–S12.

### Synthesis of azido-vancomycin; 2–4 (see Fig. 5)

*3-Azidopropane-1-amine HCl salt; S3.* 3-Bromopropylamine hydrobromide **S1** (4.00 g, 18.24 mmol) and sodium azide (5.56 g, 91.20 mmol) were added to H$_2$O (12 mL) and the reaction mixture was stirred at 75 °C for 16 h. After cooling down to room temperature, NaOH (1.00 g) was added the reaction mixture. The resulting solution was extracted with CH$_2$Cl$_2$, dried over MgSO$_4$, and concentrated under reduced pressure to yield a clear light yellow oil **S2**. To this was added aqueous HCl

to generate the hydrochloride salt (12 M, 1.5 mL). The mixture was concentrated under reduced pressure to yield a white solid **S3** (2.07 g, 82%). The residue was used for the next reaction without further purification. $^1$H NMR (600 MHz, CDCl$_3$): $\delta$ 3.39–3.37 (m, 2H), 2.82–2.80 (m, 2H), 1.76–1.71 (m, 2H); JMOD NMR (150 MHz, CDCl$_3$): $\delta$ 32.6, 39.4, 49.2. NMR data were consistent with literature[86].

*Tert-butyl (8-hydroxyoctyl)carbamate; S5.* Di-tert-butyl di-carbonate (6.18 g, 28.29 mmol) was added to a solution of 8-amino-1-octanol **S4** (4.11 g, 28.29 mmol) in a mixture of methanol/triethylamine (9:1, 90 mL). The mixture was heated to reflux for 16 h. After cooling to rt, the solvent was removed under reduced pressure and the residue was partitioned between CH$_2$Cl$_2$ and water. The organic layer was dried over MgSO$_4$ and concentrated under reduced pressure. The crude compounds were purified by MPLC over C18 silica gel (Grace Reveleris, A: H$_2$O (0.1% TFA), B: ACN (0.1% TFA), 0 → 100% B over 8 min) to give compound **S5** as a white solid (6.29 g, 91%). $^1$H NMR (600 MHz, CDCl$_3$): $\delta$ 4.48 (brs, 1H), 3.61 (t, $J = 6.6$ Hz, 2H), 3.08–3.07 (m, 2H), 1.60–1.51 (m, 4H), 1.45–1.42 (m, 11H), 1.33–1.29 (m, 6H); JMOD NMR (150 MHz, CDCl$_3$): $\delta$ 156.2, 79.2, 63.2, 40.8, 32.9, 30.2, 29.5, 29.4, 28.6, 26.9, 25.8. NMR data were consistent with literature[87].

*Tert-butyl (8-mesyloxyyoctyl)carbamate; S6.* To solution of **S5** (6.29 g, 25.64 mmol) in CH$_2$Cl$_2$ (150 mL) and triethylamine (18 mL, 128.2 mmol) was added metha-nesulfonyl chloride (3.0 mL, 38.46 mmol) at 0 °C, and the reaction mixture was stirred for 16 h. The reaction mixture was neutralised with diluted aqueous HCl (1 M), extracted with CH$_2$Cl$_2$, dried over MgSO$_4$, and concentrated under reduced pressure to obtain a white solid **S6**. The crude compound was used in the next reaction without further purification. $^1$H NMR (600 MHz, CDCl$_3$): $\delta$ 4.51 (brs, 1H), 4.22 (t, $J = 6.6$ Hz, 2H), 3.12–3.10 (m, 2H), 3.01 (s, 3H), 1.77–1.72 (m, 2H), 1.48–1.45 (m, 11H), 1.41–1.39 (m, 2H), 1.34–1.32 (m, 6H); JMOD NMR (150 MHz, CDCl$_3$): $\delta$ 156.2, 79.2, 70.3, 40.7, 37.6, 30.2, 29.3, 29.2, 29.1, 28.6, 26.8, 25.6.

*Tert-butyl (8-azidooctyl)carbamate; S7.* To a suspension of **S6** (25.64 mmol) in DMF (200 mL) was added a solution of sodium azide (4.69 g, 76.92 mmol) in water (100 mL). The reaction mixture was stirred at 110 °C for 24 h, followed by removal of the solvent under reduced pressure. The residue was dissolved in water (200 mL) extracted with CH$_2$Cl$_2$, dried over MgSO$_4$, and concentrated under reduced pressure. The crude compounds were purified by MPLC over C18 silica gel (Grace Reveleris, A: H$_2$O (0.1% TFA), B: ACN (0.1% TFA), 0 → 100% B over 8 min) to give compound **S7** as a yellow oil (4.80 g, 76%). $^1$H NMR (600 MHz, CDCl$_3$): $\delta$ 4.52 (brs, 1H), 3.25 (t, $J = 7.0$ Hz, 2H), 3.12–3.09 (m, 2H), 1.59 (tt, $J = 7.0$, 2H), 1.47–1.44 (m, 11H), 1.37–1.34 (m, 2H), 1.30 (s, 6H); JMOD NMR (150 MHz, CDCl$_3$): $\delta$ 156.2, 79.2, 51.6, 40.7, 30.2, 29.3, 29.2, 29.0, 28.6, 26.8, 26.7. NMR data were consistent with literature[88].

*8-Azidooctan-1-amine; S9.* Boc-protected compound **S7** (4.69 g, 17.35 mmol) was treated with trifluoroacetic acid (24 mL) in 78 mL CH$_2$Cl$_2$. The solution was stirred at room temperature for 2 h, neutralised with 1 M sodium hydroxide aqueous solution (400 mL), then separated and the aqueous layer extracted with CH$_2$Cl$_2$. The combined organic layer was dried over MgSO$_4$ and concentrated under

**Fig. 5 Synthesis of azido-vancomycin intermediates 2–4.** Azidoalkylamines **S3** and **S9** were synthesised via azide displacement of an activated alkylamine intermediate, with **S10** commercially available. The C-terminal carboxylic acid of vancomycin was activated using PyBOP and coupled with the azidoalkylamines to give the desired vancomycin azide intermediates.

reduced pressure. The crude compounds were purified by MPLC over C18 silica gel (Grace Reveleris, A: $H_2O$ (0.1% TFA), B: ACN (0.1% TFA), 0 → 100% B over 8 min) to give compound **S8** (2.90 g) as a clear oil. Free base **S8** was changed into the HCl salt form by adding HCl (12 M, 1.4 mL, 17.03 mmol). The mixture was concentrated under reduced pressure to yield white solid **S9** (3.51 g, 98%). [1]H NMR (600 MHz, CDCl$_3$): δ 7.87 (s, 2H), 3.26 (t, $J = 6.9$ Hz, 2H), 2.91 (t, $J = 7.6$, 2H), 1.65 (tt, $J = 7.4$, 2H), 1.59 (tt, $J = 7.1$, 2H), 1.38–1.31 (m, 8H); JMOD NMR (150 MHz, CDCl$_3$): δ 51.6, 40.0, 29.0, 28.9, 27.6, 26.7, 26.3. NMR data were consistent with literature[88].

**General procedure for synthesis of azido-vancomycin; 2–4**. Vancomycin hydrochloride (2.00 g, 1.35 mmol), PyBOP (770 mg, 1.48 mmol) in DMF (100 mL) was stirred until fully dissolved. DIPEA (1.9 mL, 10.08 mmol) was added to the reaction mixture. After 30 sec, linker (417 mg, 2.03 mmol) in DMF (4.0 mL) was immediately added to the reaction mixture. The reaction was stirred at room temperature for 16 h, and then concentrated under reduced pressure. The crude compounds were pre-purified by MPLC over C18 silica gel (Grace Reveleris, A: $H_2O$ (0.1% TFA), B: ACN (0.1% TFA), 0 → 100% B over 8 min) to give white solid **2–4**. The compounds **2–4** were repurified by Prep HPLC (flow 20 mL/min, mobile phases A = 0.1% TFA in water and B = 0.1% TFA in acetonitrile, gradient 5→100% B over 20 min) and then lyophilised to give white powder.

*Vanco-3C-N$_3$; 2*. Yield: 827 mg, 40%
   LCMS: $R_t$ = 2.69 min, @ 200 nm, [M + 2H]$^{2+}$ = 766.4, Purity by UV @ 200 nm and ELSD > 95%
   (+)-ESI-HRMS calc for C$_{69}$H$_{83}$Cl$_2$N$_{13}$O$_{23}$ [M + 2H]$^{2+}$: 765.7551, found 765.7567.
   [1]H NMR and JMOD NMR: see Figs. S13–S18 and Supplementary Data 1a.
   (+)-ESI-TOF-MS/MS: see Fig. S49.

*Vanco-8C-N$_3$; 3*. Yield: 865 mg, 45%
   LCMS: $R_t$ = 3.61 min, @ 200 nm, [M + 2H]$^{2+}$ = 801.6, Purity by UV @ 200 nm and ELSD > 95%
   (+)-ESI-HRMS calc for C$_{74}$H$_{93}$Cl$_2$N$_{13}$O$_{23}$ [M + 2H]$^{2+}$: 800.7942, found 800.7907.
   [1]H NMR and JMOD NMR: see Figs. S19–S24 and Supplementary Data 1b.
   (+)-ESI-TOF-MS/MS: see Fig. S50.

*Vanco-3PEG-N$_3$; 4*. Yield: 760 mg, 34%
   LCMS: $R_t$ = 2.9 min, @ 200 nm, [M + 3H]$^{3+}$ = 550.8, Purity by UV @ 200 nm and ELSD > 95%
   (+)-ESI-HRMS calc for C$_{74}$H$_{93}$Cl$_2$N$_{13}$O$_{26}$ [M + 2H]$^{2+}$: 824.7866, found 824.7861.
   [1]H NMR and JMOD NMR: see Figs. S25–S30 and Supplementary Data 1b.
   (+)-ESI-TOF-MS/MS: see Fig. S51.

**General procedure A for CuAAC reaction of vancomycin probes; 7–9 (see Figs. 1 and 6)**. A mixture of azide-vancomycin (1.0 eq.) and alkyne-fluorophore (1.5 eq.) were dissolved in DMF (5.0 mL), followed by addition of *t*-BuOH (5.0 mL) and $H_2O$ (5.0 mL). Aqueous CuSO$_4$ (0.2 eq. for NBD alkyne **5**, 0.5 eq. for DMACA alkyne **6**, dissolved in 0.5 mL of water), aqueous sodium ascorbate (0.4 eq. for NBD alkyne **5**, 1.0 eq. for DMACA alkyne **6**, dissolved in 0.5 mL of water), and acetic acid (10 eq. for NBD alkyne **5**, 20 eq. for DMACA alkyne **6**) were added to the reaction mixture. The reaction mixture was stirred in a microwave reactor at 100 °C for 15 min. The reaction mixture was concentrated under reduced pressure to yield the crude product. The crude compounds were pre-purified by MPLC over C18 silica gel (Grace Reveleris, A: $H_2O$ (0.1% TFA), B: ACN (0.1% TFA), 0 → 100% B over 8 min). The compounds **7–9** were repurified by Prep HPLC (flow 20 mL/min, mobile phases A = 0.1% TFA in water and B = 0.1% TFA in acetonitrile, gradient 5→100% B over 20 min) and then lyophilised to yield compounds **7–9**.

*Vanco-8C-Tz-NBD; 7*. General procedure **A**. **3** (150 mg, 9.37 × 10$^{-5}$ mol) was reacted with alkyne-NBD **5** to give an orange powder **7** (34 mg, 20%).
   LCMS: $R_t$ = 3.74 min, @ 200 nm, [M + 2H]$^{2+}$ = 607.5, Purity by UV @ 200 nm and ELSD > 95%
   (+)-ESI-HRMS calc for C$_{83}$H$_{100}$Cl$_2$N$_{17}$O$_{26}$ [M + 3H]$^{3+}$: 606.8801, found 606.8801.
   [1]H NMR and JMOD NMR: see Figs. S31–S36 and Supplementary Data 1c.
   (+)-ESI-TOF-MS/MS: see Fig. S52.

*Vanco-8C-Tz-DMACA; 8*. General procedure **A**. **3** (113 mg, 7.03 × 10$^{-5}$ mol) was reacted with alkyne-DMACA **6** to give a green powder **8** (22 mg, 16%).
   LCMS: $R_t$ = 3.71 min, @ 200 nm, [M + 2H]$^{2+}$ = 629.6, Purity by UV @ 200 nm and ELSD > 95%

**Fig. 6 Structures of vancomycin probes 7–9.** Vancomycin azides **3** and **4** were coupled with alkyne-functionalised NBD and DMACA fluorophores to give final fluorescent probes **7–9**.

(+)-ESI-HRMS calc for $C_{90}H_{110}Cl_2N_{15}O_{26}$ $[M + 3H]^{3+}$: 628.9041, found 628.9051.

[1]H NMR and JMOD NMR: see Figs. S37–S42 and Supplementary Data 1c.

(+)-ESI-TOF-MS/MS: see Fig. S53.

*Vanco-3PEG-Tz-NBD; 9.* General procedure **A**. **4** (150 mg, $9.09 \times 10^{-5}$ mol) was reacted with alkyne-NBD **5** to give an orange powder **9** (34 mg, 18%).

LCMS: $R_t = 3.24$ min, @ 200 nm, $[M + 2H]^{2+} = 623.5$, Purity by UV @ 200 nm and ELSD > 95%

(+)-ESI-HRMS calc for $C_{83}H_{100}Cl_2N_{17}O_{29}$ $[M + 3H]^{3+}$: 622.8750, found 622.8773.

[1]H NMR and JMOD NMR: see Figs. S43–S48 and Supplementary Data 1c.

(+)-ESI-TOF-MS/MS: see Fig. S54.

**Minimum inhibitory concentration (MIC) determination**. Bacteria were obtained from the American Type Culture Collection (ATCC; Manassas, VA, USA), Merck Sharp & Dohme (Kenilworth, NJ), the Coli Genetic Stock Center (CGSC, Yale University), Network on Antimicrobial Resistance in *Staphylococcus aureus* (NARSA) via BEI Resources (www.beiresources.org) and independent academic clinical isolate and mutant collections. All species used in this study were described fully in Table S4. Bacteria were cultured in Cation-Adjusted Mueller-Hinton Broth (CAMHB, Bacto Laboratories, Cat No. 212322) or Muller Hinton Broth (MHB, Bacto Laboratories, Cat No. 211443) at 37 °C overnight. A sample of each culture was then diluted 50-fold in MHB and incubated at 37 °C for 1.5–3 h. The stock

solutions of vancomycin probes **7**, **8** and **9** were prepared at 1.28 mg/mL in sterile water. The compounds were serially diluted two-fold across the wells, with concentrations ranging from 0.06 μg/mL to 128 μg/mL, plated in duplicate. The resultant mid-log phase culture was diluted to $1 \times 10^6$ CFU/mL, then 50 μL was added to each well of the compound-containing 96-well plates (Corning, Cat. No 3641, non-binding surface (NBS) plates), giving a cell density of $5 \times 10^5$ CFU/mL, and a final compound concentration range of 0.03 μg/mL to 64 μg/mL. All the plates were covered and incubated at 37 °C for 18 h with the MIC defined as the lowest compound concentration at which no bacterial growth was visible.

**Fluorescence microscopy; structured illumination microscopy (SIM)**. SIM was performed using the Zeiss Elyra PS.1 SIM/STORM microscope (green channel: laser HR Diode 488-100, filter BP495-550 + LP750; red channel: laser HR DPSS 561-100, filter BP570-620 + LP750; blue channel: laser HR Diode 405-50, filter BP420-480 + LP750). Images were analysed with ZEN2012 and Imaris for 3D-SIM. VectaShield H1000 (Abacus ALS, Cat No. VEH1000) was used as a mounting media. Cover slip glasses (Zeiss/Schott, 18 mm × 18 mm, No.1.5H) were used to prepare samples. Hank's Balanced Salt Solution (HBSS) without phenol red, CaCl₂, and MgSO₄ (Sigma Aldrich, Cat No. H6648) was used for bacterial staining. Fluorescent dyes FM4-64FX (Life Technologies, Australia, Cat No. F34653), Hoechst 33342 (Life Technologies, Australia, Cat No. H21492), and SYTO® 21 (Life Technologies, Australia, Cat No. S7556) were used for membrane staining and nucleic acid staining, respectively.

*S. aureus* ATCC 25923 was cultured in Luria broth (LB) (AMRESCO, Cat No. J106) at 37 °C overnight. A sample of each culture was then diluted 50-fold in LB

and incubated at 37 °C for 1.5–2 h. 1 mL of the resultant mid-log phase cultures were transferred to an Eppendorf tube and centrifuged (14,000 rpm, for 7 min). Bacteria were washed once with HBSS, then suspended in 20 µL of HBSS. 2 µL of this suspended bacteria solution was dropped onto a cover slip, spread and dried. An ice-cold solution (200 µL) of Hoechst 33342 (10 µg/mL in HBSS) or SYTO® 21 (2.5 µM in HBSS) was then dropped onto the bacteria, left for 5 min on ice, then drained. This was followed by adding an ice-cold solution (200 µL) of FM4-64FX (5 µg/mL in HBSS) onto the bacteria, left for 2 min on ice and then washed once with ice-cold HBSS. An ice-cold solution (200 µL) of vancomycin probes (16 µg/mL in HBSS) was then added to the bacteria, left for 30 min on ice, and then washed once with ice-cold HBSS. The bacteria were fixed with 4% paraformaldehyde 10 min for *S. aureus* on ice, followed by mounting on slides using VectaShield H1000 as a mounting media.

**SR-SIM fluorescence imaging of vancomycin probe 9 in mutant E. coli.** *E. coli* (ATCC 25922 or other strains) were cultured in LB at 37 °C overnight. A sample of each culture was then diluted 50-fold in LB and incubated at 37 °C for 1.5–2 h. 4 mL of the resultant mid-log phase cultures were transferred to an Eppendorf tube and centrifuged. Bacteria were washed once with HBSS. A solution (500 µL) of the probe **9** (32 µg/mL in HBSS) was added to an Eppendorf tube containing bacteria pellets, left for 2 h at 37 °C, and then centrifuged and washed once with HBSS. An ice-cold solution (500 µL) of FM 4-64FX (5 µg/mL in HBSS) was dropped onto the bacteria, left for 5 min on ice and then washed once with ice-cold HBSS. The pellets were suspended in 20 µL of HBSS. One µL of this suspended bacteria solution was dropped onto an agarose pad to embed bacteria for fluorescent imaging.

**Fluorescence microscopy; Airyscan confocal super-resolution microscopy.** *E. coli* (BW25113) was cultured in LB at 37 °C overnight. A sample of each culture was then diluted 1000-fold in LB and incubated at 15 °C or 37 °C for 72 h or 18 h. Then 1 mL of the cultures was harvested and washed once with HBSS. A solution (500 µL) of the probe **9** at corresponding concentrations in HBSS was added and incubated for 30 min at 37 °C with shaking. Then bacteria stained with FM4-64FX (5 µg/mL, 5 min on ice), finally, the pellets were suspended in HBSS and embedded on an agarose pad for confocal microscope Inverted LSM 880 Fast Airyscan (×63/ 1.40 OIL).

**Flow cytometric analysis of specific labelling.** Bacteria were cultured in LB at 37 °C overnight. A sample of each culture was then diluted 50-fold in LB and incubated at 37 °C for 1.5–2 h. The resultant mid-log phase cultures were harvested at 4000 rpm for 25 min, washed once with HBSS (4000 rpm, for 15 min), and resuspended in HBSS to an $OD_{600}$ of 2. Bacteria were treated with vancomycin probes at a concentration of 16 µg/mL in HBSS (37 °C, 30 min) and washed one time with 1 mL of HBSS. Bacteria cell pellets were resuspended in 1 mL of HBSS. Fluorescence intensity was measured using the flow cytometer (Gallios flow cytometer from Beckman Coulter) at a flow rate of approximately 60 µL/min, logarithmic amplification was used for the data acquisition. Ten thousand events were collected and the data was analysed using Kaluza Analysis 1.3 software. Fluorescent intensity from FL1 (Excitation 488 nm; Emission 525/20 nm) and FL9 (Excitation 405 nm; Emission 450/50 nm) were plotted against the number of events count. Positive events were gated on the histogram to estimate the mean fluorescent intensity (MFI). Total fluorescent intensity (TFI) of the gate was obtained by multiplying the MFI with the number of events on the gate.

In terms of selectivity test in a bunch of Gram-positive and Gram-negative bacteria with the fluorescent probe **9**. Bacteria were cultured in LB at 37 °C overnight. A sample of each culture was then diluted 40-fold in fresh LB and incubated at 37 °C for 1.5–2 h (OD = 0.4–0.6). The resultant mid-log phase cultures were harvested and washed once with HBSS and resuspended in HBSS to an $OD_{600}$ of 1. Then 1 mL of the cultures were transferred to Eppendorf tubes and centrifuged. A solution (500 µL) of probe **9** (32 µg/mL) was added to an Eppendorf tube containing bacteria pellets, left for 30 min at 37 °C with shaking (180 rpm). Bacteria were washed again with HBSS, then a solution (500 µL) of SYTO® 60 (5 µM in HBSS) was added and left for 10 min on ice. Bacteria were washed once with HBSS and the pellets were suspended and then diluted in HBSS for flow cytometer measurement (Cytoflex S, Beckman Coulter, Inc. 250 S. Kraemer Blvd. Brea, CA 92821 U.S.A.). The detectors used here were FITC (Excitation 488 nm; Emission 525/40 nm) and APC (Excitation 638 nm; Emission 660/10 nm). 40000 events were collected and the data was analysed using Kaluza Analysis 2.1 software (Beckman Coulter, Inc. 250 S. Kraemer Blvd. Brea, CA 92821 U.S.A.).

**Co-labelling of vancomycin probe with trimethoprim and linezolid probes.** *S. aureus* (ATCC 25923) were cultured in LB at 37 °C overnight. A sample of each culture was then diluted 50-fold in LB and incubated at 37 °C for 1.5–2 h. 1 mL of the resultant mid-log phase cultures were transferred to an Eppendorf tube and centrifuged (14,000 rpm, for 3 min). Bacteria were washed once with HBSS, then suspended in 20 µL of HBSS. 2 µL of this suspended bacteria solution was dropped onto a coverslip, spread and dried. A solution (200 µL) of trimethoprim probe **17** or linezolid probe **18** (64 µg/mL in water) was then dropped onto the bacteria, left for 30 min for probe **17** and 1 hour for probe **18**, or SYTOX® Green (2 µM, Life Technologies, Australia, Cat No. S7020), left for 5 min, then washed once with ice-

cold HBSS. This was followed by adding an ice-cold solution (200 µL) of FM4-64FX (5 µg/mL in HBSS) onto the bacteria, left for 2 min on ice, and then washed once with ice-cold HBSS. An ice-cold solution (200 µL) of vancomycin probe **9** (16 µg/mL in HBSS) was then added to the bacteria, left for 30 min on ice, and then washed once with ice-cold HBSS. The bacteria were fixed with 4% paraformaldehyde 10 min for *S. aureus* on ice, followed by mounting on slides using VectaShield H1000 as a mounting media.

**Permeability test in Gram-negative bacteria.** *E. coli* ATCC 25922 cells were cultured in LB at 37 °C overnight. A sample of each culture was then diluted 40-fold in LB and incubated at 37 °C for 1.5–2 h. The resultant mid-log phase cultures were harvested and washed once with HBSS, then resuspended in HBSS to an $OD_{600}$ of 1. Then 1 mL of the cultures were transferred to Eppendorf tubes and centrifuged. Solutions (1 mL) of antibiotics (0.125–128 µg /mL) and isopropanol (100 %) was added, then left for 1 h at 37 °C with shaking (180 rpm). Bacteria were washed again with HBSS, a solution (500 µL) of probe **9** (32 µg /mL) was added to an Eppendorf tube containing bacteria pellets, left for 30 min at 37 °C with shaking (180 rpm). Bacteria were washed again with HBSS, the pellets were suspended in 200 µL of HBSS, and 100 µL of the suspended bacteria solution was added into black 96-well plates (Black flat bottom, 96-Well plate, Corning, Cat No. CLS3915-100EA) for Tecan plate reader measurement (Tecan Plate Reader, Infinite M1000 Pro). Excitation: 475 nm; Emission: 550 nm. On the other hand, suspended bacteria solution was then diluted 1000-fold in HBSS for flow cytometer measurement (Cytoflex S, Beckman Coulter, Inc. 250 S. Kraemer Blvd. Brea, CA 92821 U.S.A.). The detector used here was FITC (Excitation 488 nm; Emission 525/40 nm). 40000 events were collected. The data was analysed using Kaluza Analysis 2.1 software (Beckman Coulter, Inc. 250 S. Kraemer Blvd. Brea, CA 92821 U.S.A.) and statistical software GraphPad Prism 8 (San Diego, CA, U.S.A.).

**Fabrication of the microfluidic devices.** The mould for the mother machine microfluidic device[89] was obtained by pouring epoxy onto a polydimethylsiloxane (PDMS, Dow Corning) replica of the original mould containing 12 independent microfluidic chips (kindly provided by S. Jun). Each of these chips is equipped with approximately 6000 lateral microfluidic channels with width and height of 1 µm each and a length of 25 µm. These lateral channels are connected to a main microfluidic chamber that is 25 and 100 µm in height and width, respectively. PDMS replicas of this device were realised as previously described[90], as summarised here. Briefly, a 10:1 (base:curing agent) PDMS mixture was cast on the mould and cured at 70 °C for 120 min in an oven. The cured PDMS was peeled from the epoxy mould and fluidic accesses were created by using a 0.75 mm biopsy punch (Harris Uni-Core, WPI). The PDMS chip was irreversibly sealed on a glass coverslip by exposing both surfaces to oxygen plasma treatment (10 s exposure to 30 W plasma power, Plasma etcher, Diener, Royal Oak, MI, USA). This treatment temporarily rendered the PDMS and glass hydrophilic, so immediately after bonding the chip was filled with 2 µL of a 50 mg/mL bovine serum albumin solution and incubated at 37 °C for 30 min, thus passivating the device's internal surfaces and preventing subsequent cell adhesion. We have also made available a step-by-step experimental protocol for the fabrication and handling of microfluidic devices for investigating the interactions between antibiotics and individual bacteria[91].

**Imaging single-cell drug accumulation dynamics.** An overnight culture was prepared as described above and typically displayed an optical density at 595 nm ($OD_{595}$) around 5. A 50 mL aliquot of the overnight culture above was centrifuged for 5 min at 4000 rpm and 37 °C. The supernatant was filtered twice (Medical Millex-GS Filter, 0.22 µm, Millipore Corp.) to remove bacterial debris from the solution and used to resuspend the bacteria in their spent LB to an $OD_{595}$ of 75. A 2 µL aliquot of this suspension was injected in the above described microfluidic device and incubated at 37 °C. The high bacterial concentration favours bacteria entering the narrow lateral channels from the main microchamber of the mother machine[92,93]. We found that an incubation time between 5 and 20 min allowed filling of the lateral channels with, typically, between one and three bacteria per channel. An average of 80% of lateral channels of the mother machine device were filled with bacteria. The microfluidic device was completed by the integration of fluorinated ethylene propylene tubing (1/32" × 0.008"). The inlet tubing was connected to the inlet reservoir which was connected to a computerised pressure-based flow control system (MFCS-4C, Fluigent). This instrumentation was controlled by MAESFLO software (Fluigent). At the end of the 20 min incubation period, the chip was mounted on an inverted microscope (IX73 Olympus, Tokyo, Japan) and the bacteria remaining in the main microchamber of the mother machine were washed into the outlet tubing and into the waste reservoir by flowing LB at 300 µL/h for 8 min and then at 100 µL/h for 2 h. Bright-field images were acquired every 20 min during this 2 h period of growth in LB. Images were collected via a ×60, 1.2 N.A. objective (UPLSAPO60XW, Olympus) and a sCMOS camera (Zyla 4.2, Andor, Belfast, UK). The region of interest of the camera was adjusted to visualise 23 lateral channels per image and images of 10 different areas of the microfluidic device were acquired at each time point in order to collect data from at least 100 individual bacteria per experiment. The device was moved by two automated stages (M-545.USC and P-545.3C7, Physik Instrumente, Karlsruhe,

Germany, for coarse and fine movements, respectively). After this initial 2 h growth period in LB, the microfluidic environment was changed by flowing minimal medium M9 with the fluorescent antibiotic probe at a concentration of 46 μg/mL at 300 μL/h for 8 min and then at 100 μL/h for 4 h. During this 4 h period of exposure to the fluorescent antibiotic derivative in use, upon acquiring each bright-field image the microscope was switched to fluorescent mode and FITC filter using custom built Labview software. A fluorescence image was acquired by exposing the bacteria to the blue excitation band of a broad-spectrum LED (CoolLED pE300white, maximal power = 200 mW Andover, UK) at 20% of its intensity (with a power associated with the beam light of 7.93 mW at the sample plane). Bright-field and fluorescence imaging during this period was carried out every 5 min. The entire assay was carried out at 37 °C in an environmental chamber (Solent Scientific, Portsmouth, UK) surrounding the microscope and microfluidics equipment.

**Image and data analysis**. Images were processed using ImageJ software as previously described[45,94] and summarised here, tracking each individual bacterium throughout the 4 h period of incubation in the vancomycin probe. A rectangle was drawn around each bacterium in each bright-field image at every time point obtaining its width, length and relative position in the hosting microfluidic channel. The same rectangle was then used in the corresponding fluorescence image to measure the mean fluorescence intensity for each bacterium that is the total fluorescence of the bacterium normalised by cell size (i.e. the area covered by each bacterium in our 2D images), to account for variations in antibiotic accumulation due to the cell cycle[85]. The same rectangle was then moved to the closest microfluidic channel that did not host any bacteria in order to measure the background fluorescence due to the presence of extracellular fluorescent antibiotic derivative in the media. This mean background fluorescence value was subtracted from the bacterium's fluorescence value. Background subtracted values smaller than 20 a.u. were set to zero since this was the typical noise value in our background measurements. All data were then analysed and plotted using GraphPad Prism 8. Statistical significance was tested using unpaired, two-tailed, Welch's $t$ test in all cases.

**Inferring single-cell kinetic parameters of antibiotic accumulation via mathematical modelling**. We modelled antibiotic accumulation using the following set of ordinary differential equations (ODEs):

$$\frac{dc(t)}{dt} = r(t) - d_c c(t)$$
$$\frac{dr(t)}{dt} = k_1 U(t - t_0) - d_r r(t) - k_2 c(t) \tag{1}$$

where $U(t - t_0)$ represents the dimensionless step function:

$$U(t - t_0) = \begin{cases} 0, & t < t_0 \\ 1, & \geq t_0 \end{cases} \tag{2}$$

Variable $c(t)$ represents the intracellular antibiotic concentration (in arbitrary units [a.u.] of fluorescence levels), and $r(t)$ [a.u./s] describes the antibiotic uptake rate. With the first equation we described how antibiotic accumulation, $c(t)$, changes over time as a result of two processes: (i) drug-uptake, which proceeds at a time-varying rate, $r(t)$; and (ii) drug loss (efflux or antibiotic transformation), which we modelled as a first order reaction with rate constant $d_c$ [s⁻¹]. With the second equation we described the dynamics of time-varying antibiotic uptake rate, $r(t)$. The uptake rate starts increasing with a characteristic time-delay (parameter $t_0$), parameter $k_1$ [a.u./s²] is the associated rate constant of this increase. We also assumed a linear dampening effect (with associated rate constant $d_r$ [s⁻¹]) to constrain the increase in uptake rate, which allowed us to recapitulate the measured saturation in antibiotic accumulation. In this model the maximum saturation is given by $F_{max} = \frac{k_1}{d_r d_c}$. Finally, the intracellular concentration has a linear negative effect on $r(t)$ (rate constant $k_2$ [a.u./s²]). We note that in this model we did not make any a priori assumptions about the mechanisms underlying antibiotic accumulation but rather aimed to capture the dynamics of the measured accumulation data.

Model parameters were inferred from single-cell fluorescence time-traces (see Image and data analysis section) using the probabilistic programming language Stan through its python interface pystan[95]. Stan provides full Bayesian parameter inference for continuous-variable models using the No-U-Turn sampler, a variant of the Hamiltonian Monte Carlo method. All No-U-Turn parameters were set to default values except parameter 'adapt_delta', which was set to 0.999 to avoid divergent runs of the algorithm. For each single-cell fluorescence time-trace we used the No-U-Turn sampler to produce 4 independent chains sampling from the parameters' posterior distribution. Each chain consisted of 1000 sampling iterations, giving in total 4000 samples from the parameters' posterior distribution. For each parameter, the median of the sampled posterior is used for subsequent analysis. For parameter inference, model time was rescaled by the length of the time-trace T, i.e. $t' = \frac{t}{T}$ so that time runs between 0 and 1, and model parameters were re-parameterised (and made dimensionless) according to the rules $(d'_c = d_c/d_r, d'_r = d_r T, k'_1 = k_1/d_r, k'_2 = k_2/d_r, t'_0 = t_0/T)$. The following diffuse priors were used for the dimensionless parameters, where $U(a, b)$ denotes the uniform distribution in the range $[a, b]$: $d'_c \sim U(0, 1)$ so that uptake rate dynamics

are always faster than drug-accumulation dynamics, i.e., $d_c < d_r$; $\log_{10} d'_r \sim U(0, 3)$ constraining the timescale associated with $d_r$ to be shorter than the timescale of the experiment, i.e., $1/d_r < T$; $\log_{10} k'_1 \sim U(0, 3)$ and $\log_{10} k'_2 \sim U(-3, 0)$, so that the parameter controlling adaptive inhibition is small enough and there is no oscillatory behaviour in the model i.e., $k_2 < k_1$; $t'_0 \sim U(0, 1)$, since the transformed time $t'$ runs from 0 to 1.

**Statistics and reproducibility**. The statistical tests and number of biological replicates and/or experiments are stated in the figure subtexts. Statistical analysis was done using Graphpad Prism 8.

**Reporting summary**. Further information on research design is available in the Nature Portfolio Reporting Summary linked to this article.

## Data availability
All relevant data are available in this article, its Supplementary Information and Supplementary Data files (the source data behind the graphs in the paper is contained in Supplementary Data 2), except for original image files, which are available from the corresponding author upon reasonable request.

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

## Acknowledgements
B.Z. was supported by a China Scholarship Council (CSC) Scholarship, University of Queensland Research Training Tuition Fee Offset and a Research Higher Degree Top Up Scholarship. W.P. by a UQ International Scholarship (UQI) and IMB Post-graduate Award (IMBPA), and M.R.L.S. by an Australian Postgraduate Award and an Institute for Molecular Biosciences Research Advancement Award. U.L. was supported through a BBSRC responsive mode grant (BB/V008021/1), an MRC Proximity to Discovery EXCITEME2 grant (MCPC17189) and an award from the Gordon and Betty Moore Foundation Marine Microbiology Initiative (GBMF5514). K.T.-A. gratefully acknowledges financial support from the EPSRC via grant EP/T017856/1.

The project has been funded in part by Wellcome Trust Strategic Grant WT1104797/Z/14/Z, NHMRC Development grant APP1113719 and NHMRC Project/Ideas grants APP2004367 and APP1026922. This work was further supported by a Royal Society Research Grant (RG180007) awarded to S.P., a QUEX Initiator grant awarded to S.P., K.T.-A. and M.A.T.B., and a GW4 Initiator award to K.T.-A. and S.P. Microscopy was performed at the Australian Cancer Research Foundation (ACRF)/Institute for Molecular Bioscience Cancer Biology Imaging Facility, which was established with the support of the ACRF. The mutant *tolC*, *lpxC* and DC2 *E. coli* strains MB4902, MB5746, MB5747 were generously supplied by Merck Sharp & Dohme (Kenilworth, NJ)[96]. CFT073 and *waaL* mutants were generously supplied by Professor Mark Schembri (School of Chemistry and Molecular Biology - University of Queensland), and the *E. coli* strain BW25113 by Professor Ian Henderson (Institute for Molecular Bioscience - University of Queensland). Gram-positive clinical isolates documented in source as "Clinical isolate - Australia" were kindly provided by Professor David Paterson. (University of Queensland Centre for Clinical Research).

## Author contributions
M.A.T.B. designed the project. W.P., B.Z. and M.A.T.B. wrote the manuscript with input and review from all authors; M.A.T.B., M.A.C., M.S.B., W.P., M.R.L.S., S.K. and B.Z. designed and coordinated experiments and analysed results, W.P. and M.R.L.S., designed, synthesised, purified and analysed compounds, and W.P., M.R.L.S., and B.Z. carried out the in vitro microbiological assays, confocal microscopy and flow cytometry. U.L. and S.P carried out the single-cell microfluidics-microscopy experiments. M.V. and K.T.-A. developed and implemented the mathematical model. All authors discussed the results and commented on the manuscript.

## Competing interests
The authors declare the following competing interests: M.S.B., M.R.L.S., W.P., M.A.C. and M.A.T.B. are inventors on WO/2018/102890 "Visualisation Constructs", and W.P., M.A.C. and M.A.T.B. are inventors on "WO/2018/102889 "Glycopeptide Antibiotic Constructs", which include compounds described in this publication, which may be subject to commercialisation. The remaining authors declare no competing interests.
