## [Peer Review File · Communications Biology]

Reviewers' comments:

Reviewer #1 (Remarks to the Author):

The authors describe the synthesis and subsequent in vitro testing of vancomycin fluorescent probes, with small fluorophores and minimal electronic charge to maintain their antimicrobial activity, which can readily be used in standard methods, including flow cytometry. The straightforward synthesis involves the modification of vancomycin with three different azide linkers which, in the second step, are used to couple NBD and DMACA as fluorophores via copper-catalyzed azide-alkyne cycloaddition. Three linker-fluorophore combinations were chosen, based on their chemical properties and antimicrobial activity, for further investigation. They showed that the chosen fluorescent probes selectively stain Gram-positive bacteria. Super resolution microscopy revealed the binding of the probes to the newly synthesized peptidoglycan layer at the dividing septum. Interestingly one probe could be used to indicate membrane damage in Gram-negative bacteria, induced by different stressors, making it a promising tool in investigating the mode of action of novel antibiotic substances.

The paper is overall very well written and provides the reader with all necessary background-, synthesis- and assay-specific information. The authors create a broad but compact overview on the current state of the research area in their introduction. They include approaches of other research groups, while highlighting the need for further tools to study the complex interactions and functions in bacterial systems, to combat the emerging threat of antibiotic-resistance. The facile synthesis of the presented vancomycin-fluorophore conjugates through azide-alkyne cycloaddition provides a robust synthesis route with sufficient yields. Specifically, vancomycin probe 9 has potential to be a great tool to assess the effect of antibiotic substances on the membrane of Gram-negative bacteria. Since identifying the mode of action of antibiotics is a challenging and time-consuming task, this fluorescent probe could find use for many researchers in the field. In my opinion the presented paper needs no further improvements.

Reviewer #2 (Remarks to the Author):

The manuscript describes a comprehensive synthesis and evaluation of probes of vancomycin. Overall the manuscript was suitably put together, reads well and is technically sound. In the reviewers opinion, the manuscript may make a modest impact on the field but having additional imaging probes for vancomycin and strategies to synthesize them is valuable.

Major criticism of the work is as follows.

1. The authors state that there is a need for better probes that have antibacterial activity; also citing BODIPY VAN as one of these probes where the technology described would be better. However, it would have been compelling and scientifically rigorous to include BODIPY VAN in at least some of the experiments to support the claim made in the study.

Minor comments.

Figure 1c could be better reflected as a bar chart.

Are there any limitations to the study?

Reviewer #3 (Remarks to the Author):

In this manuscript, the authors reported a type of vancomycin fluorescent probes and investigated their application in detection of bacteria. The results showed that these fluorescent probes can be used to detect the bacteria by plate reader quantification, flow cytometry analysis, high-resolution microscopy imaging, and single cell microfluidics analysis. This work presented some interesting results and almost all experiments supported the authors' claim. So this reviewer is of the opinion that

this manuscript can be accepted to be published in Communications Biology only after major revision.

- 1) For such probes containing many peptide moieties, how about the stability of these probes? How to keep their detecting activity?
- 2) These probes can efficiently distinguish Gram-positive and Gram-negative bacteria, which is a very common behavior like most fluorescent probes. What exactly are the advantages of these probes?
- 3) From molecular structures of these probes, they ought to be fluorescent. How do they show their responsiveness to bacteria?

Response to Reviewers

Reviewer #1 (Remarks to the Author):

The authors describe the synthesis and subsequent in vitro testing of vancomycin fluorescent probes, with small fluorophores and minimal electronic charge to maintain their antimicrobial activity, which can readily be used in standard methods, including flow cytometry. The straightforward synthesis involves the modification of vancomycin with three different azide linkers which, in the second step, are used to couple NBD and DMACA as fluorophores via copper-catalyzed azide-alkyne cycloaddition. Three linker-fluorophore combinations were chosen, based on their chemical properties and antimicrobial activity, for further investigation. They showed that the chosen fluorescent probes selectively stain Gram-positive bacteria. Super resolution microscopy revealed the binding of the probes to the newly synthesized peptidoglycan layer at the dividing septum. Interestingly one probe could be used to indicate membrane damage in Gram-negative bacteria, induced by different stressors, making it a promising tool in investigating the mode of action of novel antibiotic substances.

The paper is overall very well written and provides the reader with all necessary background-, synthesis- and assay-specific information. The authors create a broad but compact overview on the current state of the research area in their introduction. They include approaches of other research groups, while highlighting the need for further tools to study the complex interactions and functions in bacterial systems, to combat the emerging threat of antibiotic-resistance. The facile synthesis of the presented vancomycin-fluorophore conjugates through azide-alkyne cycloaddition provides a robust synthesis route with sufficient yields. Specifically, vancomycin probe 9 has potential be a great tool to assess the effect of antibiotic substances on the membrane of Gram-negative bacteria. Since identifying the mode of action of antibiotics is a challenging and time-consuming task, this fluorescent probe could find use for many researchers in the field. In my opinion the presented paper needs no further improvements.

Response: We appreciate the positive feedback from the reviewer.

Reviewer #2 (Remarks to the Author):

The manuscript describes a comprehensive synthesis and evaluation of probes of vancomycin. Overall the manuscript was suitably put together, reads well and is technically sound. In the reviewers opinion, the manuscript may make a modest impact on the field but having additional imaging probes for vancomycin and strategies to synthesize them is valuable.

Major criticism of the work is as follows.

1. The authors state that there is a need for better probes that have antibacterial activity; also citing BODIPY VAN as one of these probes where the technology described would be better. However, it would have been compelling and scientifically rigorous to include BODIPY VAN in at least some of the experiments to support the claim made in the study.

Response: We have now compared the antimicrobial activity of probes **7**, **8** and **9** to the commercially available vancomycin-derived fluorescent probes, Van-FITC (SBR00028, Sigma-Aldrich), and Van-BODIPY (V34850, Invitrogen™). Compared to Van-BODIPY, Probe **7** retained two-fold improved activity against all tested *S. aureus* strains, whereas Van-FITC did not possess antimicrobial activity at the highest concentration tested (32 µg/mL) (**Table S3**). For testing of Van-FITC and Van-BODIPY, they were both dissolved in DMSO according to their product manuals, instead of the water used for probes 7–9. Given that water is less disruptive to bacterial survival than DMSO, the ability to dissolve the new probes in water should cause less experimental interference.

We also compared the staining ability of probe **7** and probe **9** to the commercial fluorescent probes using flow cytometry. The Van-BODIPY derivative provided the strongest staining intensity of *S. aureus* under the tested conditions. Probes **7** and **9** had intermediate intensity, while the staining ability of Van-FITC was the worst (**Figure S4A**). However, when incubated with the *E. coli* cells, the Van-BODIPY probe stained a larger population of cells than the other probes, especially at 2 µg/mL, indicating a degree of non-specificity. In contrast, Probes **9** and Van-FITC did not show much staining with *E. coli* at either concentration, while Probe **7** was similar to Probe **9** at 2 µg/mL but had the highest *E. coli* labelling at 16 µg/mL (**Figure S4B**).

Airyscan confocal super-resolution imaging was used to compare vancomycin probes **7** and **9** against the commercial probes Van-FITC and Van-BODIPY for staining of *S. aureus* ATCC 25923. With the exception of Van-FITC, all fluorescent probes showed good peptidoglycan binding ability. However, while Probes **7** and **9** showed very strong fluorescence at the cell division septum (the site of Lipid II precursor incorporation into peptidoglycan) compared to the lateral wall, the Van-BODIPY probe appeared to label the cells more uniformly, potentially indicating reduced selectivity of binding to nascent peptidoglycan. More intracellular labelling was also observed with the Van-BODIPY probe, further supporting that Probes **7** and **9** were more specific than Van-BODIPY (**Figure S4C**). Van-FITC did not show good staining ability, consistent with its poor MIC and flow cytometry staining, with no fluorescence signal observed at 2 µg/mL and less clear labelling than other tested probes at the higher concentration (**Figure S4C**).

Minor comments.

Figure 1c could be better reflected as a bar chart.

Response: We believe the reviewer is referring to Figure 2C. We respectfully argue that the current line chart is more effective at showing the variation in labelling with concentration for the different probes.

Are there any limitations to the study?

Response: Although we have generated green (NBD) and blue (DMACA) versions of the probes, it is still worth developing fluorescent vancomycin probes with other colours in order to be more flexible when applying them in conjunction with other fluorescent probes, which is a future direction of this study.

While we have tested the probes against a range of Gram-positive and negative bacteria, there are still many other species that could be tested, which might give differing staining/selectivity.

Reviewer #3 (Remarks to the Author):

In this manuscript, the authors reported a type of vancomycin fluorescent probes and investigated their application in detection of bacteria. The results showed that these fluorescent probes can be used to detect the bacteria by plate reader quantification, flow cytometry analysis, high-resolution microscopy imaging, and single cell microfluidics analysis. This work presented some interesting results and almost all experiments supported the authors' claim. So this reviewer is of the opinion that this manuscript can be accepted to be published in Communications Biology only after major revision.

1) For such probes containing many peptide moieties, how about the stability of these probes? How to keep their detecting activity?

Response: While peptidic, the vancomycin core of the probes is known to possess good stability. The chemical stability of Probe **9** was assessed by analytical testing (HPLC) of old stock solutions that were dissolved in ddH₂O and stored at -20 °C for over 3 years. No significant degradation was evident, with purity still > 95% (data not shown). Furthermore, a one-year-old stock of Probe **9** showed identical antimicrobial activity as the fresh stock, indicating that the probe can be used for at least one year once dissolved (**Table S3**).

We also compared the staining ability of one-year-old stock of Probe **9** to the fresh stock using flow cytometry. They performed quite similar in staining *S. aureus* with good fluorescent intensity under the tested conditions. Similarly, when incubated with the *E. coli* cells, the old stock and fresh stock of Probe **9** behaved similarly, and showed minimal staining of *E. coli* at either concentration (**Figure S4B**).

Finally, Airyscan confocal super-resolution imaging results also showed that one-year-old stock of Probe **9** strongly stained *S. aureus* ATCC 25923 in a similar manner to the fresh stock (**Figure S4C**).

To summarize, our probes are quite stable when stored at -20 °C for over 3 years and we have demonstrated that one-year-old stock works effectively under experimental conditions.

2) These probes can efficiently distinguish Gram-positive and Gram-negative bacteria, which is a very common behavior like most fluorescent probes. What exactly are the advantages of these probes?

Response: Traditionally, Gram-positive and Gram-negative bacteria are distinguished by Gram-staining, which relies on uptake and retention of a crystal violet stain to label the thick peptidoglycan layer of Gram-positive bacteria. However, many factors including the intactness and size of bacterial cells will affect the staining accuracy of this commonly used technique and it is not suitable for investigating viable bacterial cells because of a fixation step. Other, non-specific fluorophores, would behave similarly. Our vancomycin-derived fluorescent probes are able to easily distinguish Gram-positive and Gram-negative bacteria among viable cells. In addition, compared to other non-specific fluorescent probes, our probes display similar antimicrobial profile as the parent antibiotic vancomycin, and thus can also be used to study the mode of action of vancomycin. Importantly, their use to measure of outer-membrane permeabilisation of Gram-negative bacteria is based on the specific mechanism of binding to peptidoglycan exposed by a leaky membrane, which provides a greater level of rigor than probes which might accumulate intracellularly.

These mechanism-specific fluorescent probes could become a useful addition to the repertoire of tests commonly used to help define the mode of action of new antimicrobial compounds and the effects of bacterial pathway mutations.

3) Form molecular structures of these probes, they ought to be fluorescent. How do they show their responsiveness to bacteria?

Response: We are unclear of exactly what is being asked, as the structures of the probes include the NBD fluorophore which is designed to be fluorescent – the fluorescence signal does not depend on interactions with the bacteria. We have tested the NBD fluorophore alone as a control, and it did not show any labelling of the bacteria, confirming that the antimicrobial activity and labelling was due to the incorporation of the vancomycin main structure itself. Thus, the fluorescent vancomycin can be used to mimic the mode of action of the parent vancomycin.

REVIEWERS' COMMENTS:

Reviewer #2 (Remarks to the Author):

The authors have adequately addressed comments raised and have objectively improved the quality of the paper with new data concerning commercially available probes. The new probes made in the paper and comparison with existing probes will provide the field with new tools from which to chose for examination of bacterial cell wall.

Reviewer #3 (Remarks to the Author):

The authors have addressed all issues I raised. This revised manuscript is now acceptable for publication in Communication Biology.